# Targeted Low-rank Refinement:
# Enhancing Sparse Language Models with Precision

Li Shen [1]  Anke Tang [† 2]  Yong Luo [† 2]  Tao Sun [3]  Han Hu [4]  Xiaochun Cao [1 5]

## Abstract

Pruning is a widely used technique for compressing large neural networks that eliminates weights with minimal impact on performance. Current pruning methods, exemplified by magnitude pruning, assign importance scores to weights based on their magnitude and remove those below a certain threshold. However, these methods introduce a gap between the original dense and pruned sparse models, potentially impairing performance, especially at high sparsity ratios. To address this issue, we introduce a method that bridges this gap through low-rank approximation of the difference between dense and sparse matrices. Our approach iteratively refines the sparse weight matrix with a low-rank adjustment, capturing essential information typically lost during pruning. We provide a comprehensive theoretical analysis of our method, establishing its convergence properties and efficacy. Experimental results on LLaMA models validate our method's effectiveness across various pruning techniques and sparsity levels. At 50% sparsity, it reduces perplexity by 53.9% compared to conventional magnitude pruning on LLaMA-7B. Furthermore, our approach enables an 8.6% reduction in model parameters while maintaining a sparsity ratio of about 50%.

[1]School of Cyber Science and Technology, Shenzhen Campus of Sun Yat-sen University, Shenzhen 518107, China [2]National Engineering Research Center for Multimedia Software, School of Computer Science and Hubei Key Laboratory of Multimedia and Network Communication Engineering, Wuhan University, Wuhan 430072, Hubei, China [3]National University of Defense Technology, Hunan, China [4]School of Information and Electronics, Beijing Institute of Technology, Beijing, China [5]Key Laboratory of Cyberspace Security, Ministry of Education, China. Correspondence to: Anke Tang <anketang@whu.edu.cn>, Yong Luo <luoyong@whu.edu.cn>.

*Proceedings of the 42nd International Conference on Machine Learning*, Vancouver, Canada. PMLR 267, 2025. Copyright 2025 by the author(s).

## 1. Introduction

Pruning is a crucial technique in the field of model compression, particularly for large language models (LLMs), which have become the cornerstone of natural language processing tasks (Devlin, 2018; Brown, 2020; Hoffmann et al., 2022). Pruning involves the removal of specific weights or parameters from the neural network that are considered to have minimal impact on the overall performance of the model (LeCun et al., 1989; Hassibi et al., 1993; Han et al., 2015b; Frankle & Carbin, 2018; Frankle et al., 2020). This technique is especially valuable for LLMs, which often contain billions of parameters and require substantial computational resources for both training and inference (Han, 2017; Touvron et al., 2023; Minaee et al., 2024).

One of the most representative pruning techniques is magnitude pruning, which removes weights that have the smallest absolute values. This method is based on the assumption that smaller weights have less effect on the network's overall performance (Han et al., 2015a). Furthermore, as model sizes continue to grow, the number of redundant parameters also increases. For LLMs with billions of parameters, even half of the layers can be dropped without significantly affecting performance (Men et al., 2024; Fan et al., 2024). However, recent research shows that pruning can cause irreparable loss of knowledge and performance drops, especially for difficult tasks, a phenomenon known as the *Junk DNA Hypothesis* (Yin et al., 2024). This consistent degradation in performance is observed across a spectrum of pruning methods, including magnitude pruning, SparseGPT (Frantar & Alistarh, 2023), and Wanda (Sun et al., 2023), and applies to both unstructured pruning and structured N:M pruning.

Conventional approaches to post-pruning performance recovery face a three-fold challenge: (1) *Computational burden*: Re-training (Frankle & Carbin, 2018; Xia et al., 2023; Kim et al., 2024a) and knowledge distillation (Hinton, 2015; Wan et al., 2024a; Muralidharan et al., 2024) methods are computationally expensive and time-consuming. (2) *Data and model dependency*: These techniques typically require either extensive datasets or access to a high-performing teacher model, which may not always be feasible. (3) *Sparsity inconsistency*: Recent low-rank approximation methods (Li et al., 2023b; Mozaffari et al., 2024; Zhang & Pa-

pyan, 2024) often fail to maintain a consistent sparsity pattern, making them unsuitable for structured pruning, which is crucial for hardware efficiency.

In this study, we address these challenges of post-pruning performance recovery by approximating the dense matrix as the sum of an updated sparse matrix with a maintained sparsity pattern and a low-rank matrix. We propose an iterative refinement process that concurrently updates the sparse matrix and the low-rank component in a data-free manner. This approach effectively recovers crucial information typically lost during pruning. Our method features an adaptive low-rank approximation that dynamically adjusts to complement the sparse matrix, enabling efficient information recovery. Unlike traditional techniques that rely on large datasets or high-performing teacher models, our approach operates directly on model weights, offering a computationally efficient and broadly applicable solution. This approach aims to improve the performance of the pruned model without significantly increasing the parameter count. Our method combines the advantages of both sparse and low-rank approximation, ensuring the model maintains its efficiency while enhancing its accuracy. In addition to our empirical findings, we provide a comprehensive theoretical analysis of the iterative refinement process, which rigorously demonstrates the favorable convergence properties of our method.

Experimental results on the LLaMA models demonstrate the effectiveness of our low-rank refinement approach. When applying 50% sparsity, our method achieves a 53.9% reduction in perplexity compared to conventional magnitude pruning. The benefits of our approach become increasingly evident as sparsity increases: at 60% sparsity, we observe a 92.0% decrease in perplexity compared to vanilla sparse models, while at 70% sparsity, an impressive 99.6% reduction is achieved. This highlights the effectiveness of our method, especially in scenarios of high sparsity where traditional approaches typically face considerable challenges.

To summarize, the main contributions of this paper are:

- We bridge the gap between the original dense and pruned sparse model by leveraging a low-rank component. This approach effectively fills the gap left by pruning, enhancing the model's performance with minimal parameter increase.

- We develop an iterative algorithm that incrementally refines the sparse weight matrix and incorporates the low-rank approximation. By prioritizing the preservation of weight components associated with larger singular values, our method allows for a more aggressive reduction of less important components, leading to a more precise approximation.

- We provide a thorough theoretical analysis of our proposed method, which offers a rigorous foundation understanding of the effectiveness, convergence, and stability of our approach.

- We evaluate our method on LLMs and show significant perplexity improvements over baselines. This is particularly noteworthy at high sparsity levels, maintaining robust improvements even as high as 70% sparsity.

## 2. Related Work

**LLM compression.** LLMs have become essential in natural language processing tasks, but their substantial size poses challenges in terms of computational resources and efficiency. Various techniques have been proposed to compress LLMs while maintaining their performance, including pruning, low-rank compression, quantization, and knowledge distillation (Cheng et al., 2017; Choudhary et al., 2020; Haroush et al., 2020). In this study, we concentrate on pruning methods and use low-rank approximations to restore the lost performance. Pruning LLMs with billions of parameters differs significantly from pruning smaller models (Gale et al., 2019; Frankle et al., 2020; Kurtic & Alistarh, 2022), as current pruning techniques often necessitate extensive re-training after the pruning process, which is prohibitively costly for LLMs (Komatsuzaki et al., 2022; Chung et al., 2024; Snell et al., 2024). Pruning methods are often categorized into unstructured and structured pruning (Liu et al., 2017; Fan et al., 2019; He & Xiao, 2023). Structured pruning techniques, including layer pruning (Chen & Zhao, 2018; Kim et al., 2024b), channel pruning (He et al., 2017; Zhuang et al., 2018), and N:M pruning (Sun et al., 2021), focus on eliminating entire neurons, layers, or N out of M elements in a regular pattern. Unstructured pruning removes individual weights without considering their structure, often improving performance but making it less efficient for hardware. Low-rank compression reduces the dimensions of the weight matrix by focusing on larger singular values in both the column and row spaces (Cheng et al., 2005; Idelbayev & Carreira-Perpinán, 2020). While quantization (Xiao et al., 2023; Lin et al., 2024) and knowledge distillation (Wan et al., 2024b) have been used to compress LLMs, they are orthogonal to our approach. Low-rank refinement is also used to regain performance during the quantization process (Guo et al., 2023; Loeschcke et al., 2024; Li et al., 2023a).

**Post-pruning performance recovery.** To recover the performance after pruning step, several post-pruning recovery techniques have been explored such as re-training the pruned model and using knowledge distillation (Muralidharan et al., 2024). However, these approaches can be computationally expensive and time-consuming. Recent research has focused on incorporating low-rank approximations to recover the lost performance. By adding a low-rank component to the pruned model, it is possible to approximate the original dense model more closely with minimal computational

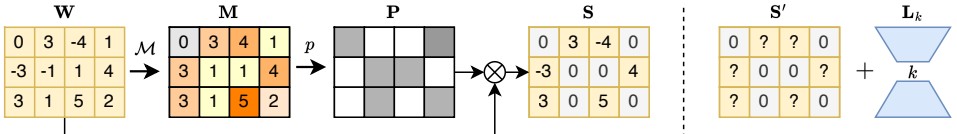

(a) The general framework of neural network pruning.  (b) Refine with a low-rank patch.

Figure 1: (a) An overview of the general framework of (local) layer-wise pruning, where $\boldsymbol{W}$, $\boldsymbol{M}$, $\boldsymbol{P}$, and $\boldsymbol{S}$ are the original dense weight matrix, the importance score matrix, the binary pruning mask, and the pruned sparse weight matrix, respectively. Here we show the case of pruning using the magnitude of the weights as the importance score. (b) Our proposed method, where $\boldsymbol{S}$ is updated and a low-rank matrix $\boldsymbol{L}_k$ is introduced to minimize the approximation error.

overhead (Chandrasekaran et al., 2009; Li et al., 2023b; Mozaffari et al., 2024; Zhang & Papyan, 2024).

**Low-rank refinement.** Low-rank refinement is a technique that has been used to recover the performance in many scenarios. While this study concentrates on post-pruning performance recovery, low-rank refinement has also been utilized in other scenarios, including quantization (Guo et al., 2023; Loeschcke et al., 2024; Li et al., 2023a; Zhang et al., 2023). However, a key distinction of our approach is that it is data-free and can be applied to individual layers, making the refinement process more memory-efficient.

## 3. Preliminary

### 3.1. The General Framework of Layer-Wise Pruning

Neural network pruning is a crucial technique for model compression, aiming to reduce parameters while preserving performance. In this section, we first present the general framework for (local) layer-wise pruning, outlining the key steps and components involved as illustrated in Fig. 1(a). Let $\boldsymbol{W} \in \mathbb{R}^{m \times n}$ represent a weight matrix, and $\boldsymbol{M} = \mathcal{M}(\boldsymbol{W}, \mathcal{D}) \in \mathbb{R}^{m \times n}$ denote its corresponding importance score matrix. Here, $\mathcal{M} : \mathbb{R}^{m \times n} \times \mathcal{D} \to \mathbb{R}^{m \times n}$ is a metric function that computes the importance scores based on the weight matrix $\boldsymbol{W}$ and an optional dataset $\mathcal{D}$. The most straightforward approach is to use the magnitude of the weights as the importance score, i.e., $\boldsymbol{M}_{ij} = |\boldsymbol{W}_{ij}|$. More sophisticated methods can be employed to capture the importance of each weight more accurately, such as the SparseGPT (Frantar & Alistarh, 2023) and Wanda (Sun et al., 2023). Given a pruning ratio $p \in [0, 1]$, we set threshold $h$ as the $p$-th percentile of $\boldsymbol{M}$ as the decision boundary for pruning. Then the binary pruning mask $\boldsymbol{P}$ is obtained by $\boldsymbol{P}_{ij} = \mathbb{I}(\boldsymbol{M}_{ij} > h)$. The pruned sparse weight matrix $\boldsymbol{S}$ is obtained by $\boldsymbol{S} = \boldsymbol{W} \odot \boldsymbol{P}$, where $\odot$ denotes element-wise multiplication. This zeroes out less important weights while retaining significant ones. An optional weight update procedure can be applied to refine the pruned weights further. For example, after the pruning process, a re-training phase may be implemented to fine-tune the remaining weights and

recover some lost performance. During this process, the pruning structure is maintained by enforcing the constraint $\boldsymbol{S}^{(t)} = \boldsymbol{S}^{(t)} \odot \boldsymbol{P}$ at each iteration $t$. This ensures that the pruned weights remain zero throughout the re-training process, preserving the sparsity achieved through pruning.

### 3.2. Singular Value Decomposition

Here we briefly review the Singular Value Decomposition (SVD) of a matrix. Given a matrix $\boldsymbol{W} \in \mathbb{R}^{m \times n}$, its reduced SVD is given by (Olver & Shakiban, 2018):

$$\boldsymbol{W} = \boldsymbol{U}\boldsymbol{\Sigma}\boldsymbol{V}^{\top} = \sum_{i=1}^{r} \sigma_i \boldsymbol{u}_i \boldsymbol{v}_i^{\top}, \tag{1}$$

where $\boldsymbol{U} \in \mathbb{R}^{m \times r}$ and $\boldsymbol{V} \in \mathbb{R}^{r \times n}$ are orthogonal matrices, and $\boldsymbol{\Sigma} \in \mathbb{R}^{r \times r}$ is a diagonal matrix containing the singular values of $\boldsymbol{W}$. The $i$-th columns of matrices $\boldsymbol{U}$ and $\boldsymbol{V}$ are represented by $\boldsymbol{u}_i$ and $\boldsymbol{v}_i$, respectively. Additionally, we use $\sigma_i$ to denote the $i$-th diagonal element of the matrix $\boldsymbol{\Sigma}$. For convenience, we extend this notation to represent functions that map $\boldsymbol{W}$ to its SVD components. This allows us to express the decomposition as $\boldsymbol{W} = \boldsymbol{U}(\boldsymbol{W})\boldsymbol{\Sigma}(\boldsymbol{W})\boldsymbol{V}^{\top}(\boldsymbol{W})$, or alternatively as a sum of outer products: $\boldsymbol{W} = \sum_{i=1}^{r} \sigma_i(\boldsymbol{W})\boldsymbol{u}_i(\boldsymbol{W})\boldsymbol{v}_i^{\top}(\boldsymbol{W})$. While this slightly abuses the original notation, it provides a concise way to refer to the SVD components of a matrix.

## 4. Method

Low-rank approximation is a common technique for model compression, which has been widely used in various fields as in (Zhou & Tao, 2011; Yu et al., 2017). In this study, we focus on the local pruning method and aim to fill the gap between the original dense weight matrix $\boldsymbol{W}$ and the pruned weight $\boldsymbol{S}$ by introducing a low-rank matrix $\boldsymbol{L}_k$ with a target rank $k$ to approximate $\Delta \boldsymbol{W} = \boldsymbol{W} - \boldsymbol{S}$. We illustrate the main idea in Fig. 1(b), where the dense matrix $\boldsymbol{W}$ is factorized into the superposition of a low-rank matrix $\boldsymbol{L}$ and a updated sparse matrix $\boldsymbol{S}'$:

$$\boldsymbol{W} = \boldsymbol{S}' + \boldsymbol{L} \approx \boldsymbol{S}' + \boldsymbol{L}_k. \tag{2}$$

**Algorithm 1** The Proposed Iterative Weight Update Method

1: **Inputs:** Dense weight matrix $W$, binary mask $P$, target rank $k$, number of iterations $T$
2: Initialize $S^{(0)} \leftarrow W \odot P$
3: **for** $t = 0$ to $T - 1$ **do**
4: $\quad L^{(t)} \leftarrow W - S^{(t)}$
5: $\quad$ Compute SVD: $L^{(t)} = U^{(t)} \Sigma^{(t)} V^{(t)\top}$
6: $\quad r^{(t)} \leftarrow \lfloor 1 + \frac{k-1}{T-1} t \rfloor$
7: $\quad S^{(t+1)} \leftarrow S^{(t)} + P \odot \left\{ U^{(t)}_{r^{(t)}:} \Sigma^{(t)}_{r^{(t)}:} V^{(t)\top}_{r^{(t)}:} \right\}$
8: **end for**
9: $L^{(T)} \leftarrow W - S^{(T)}$
10: **Returns:** $S^{(T)}, L^{(T)}$

---

In this formulation, $S'$ should maintain the same sparsity pattern as $S$, ensuring that $S' = S' \odot P$ and $S = S \odot P$ are satisfied. It's important to note that both $S'$ and $L$ can have elements of any magnitude without restrictions. $L_k$ is the best rank-$k$ approximation of $L$, which can be obtained using the SVD of $L$ as $L_k = U_{:k}(L)\Sigma_{:k}(L)V_{:k}^\top(L)$. Beginning with fixed $S$, we have the baseline:

*Baseline* 1 (Zero-shot SVD). The most straightforward method to obtain the low-rank matrix $L_k$ is to directly perform the SVD on difference matrix $W - S$ without updating $S$. This can be expressed as $L_k = U_{:k}(W - S)\Sigma_{:k}(W - S)V_{:k}^\top(W - S)$. Here, $U_{:k}(\cdot)$, $\Sigma_{:k}(\cdot)$, and $V_{:k}(\cdot)$ represent the first $k$ columns of $U(\cdot)$, the top-left $k \times k$ submatrix of $\Sigma(\cdot)$, and the first $k$ columns of $V(\cdot)$, respectively.

**Parameter efficiency analysis.** We analyze the computational efficiency of low-rank refinement by examining its parameter count and FLOPs (floating-point operations). We compare these metrics with the dense model and a pruned model, considering the impact of sparsity. For a weight matrix $W \in \mathbb{R}^{m \times n}$, the dense model has $mn$ parameters and requires $2mn$ FLOPs for a forward pass. With pruning at sparsity ratio $p$, these reduce to $(1 - p)mn$ parameters and $2(1 - p)mn$ FLOPs. Our method introduces a low-rank matrix $L_k = BA$, which introduces $k(m + n)$ parameters and $2k(m + n)$ FLOPs, slightly reducing the overall sparsity ratio by $k(\frac{1}{m} + \frac{1}{n})$. The choice of $k$ and $p$ presents a trade-off between model size, computational complexity, and performance. In practice, using a rank $k$ of 128 results in only a 4.9% increase in parameters, while possibly reducing perplexity by half on a LLaMA-7B model with both unstructured 50% sparsity and structured 4:8 sparsity.

This optimization problem is generally NP-hard and similar challenges have been addressed in the fields of matrix completion (Chandrasekaran et al., 2011) and robust Principal Component Analysis (PCA) (Candès et al., 2011; Peng et al., 2020) by solving a Principal Component Pursuit (PCP) as:

$$\min_{L,S'} \|L\|_* + \lambda \|S'\|_1, \quad \text{s.t. } L + S' = W. \quad (3)$$

Where $\|\cdot\|_*$ denotes the nuclear norm and $\|\cdot\|_1$ denotes the $\ell_1$-norm, $\lambda$ is a hyperparameter that controls the trade-off between the rank of $L$ and the sparsity of $S'$. The nuclear norm serves as a convex approximation for the rank of a matrix, while the $\ell_1$-norm acts as a convex proxy for the $\ell_0$-norm (which represents the count of non-zero elements in a matrix). In this convex optimization problem, the $\|L\|_*$ term promotes a low-rank solution for $L$, whereas the $\lambda\|S'\|_1$ term promotes sparsity in $S'$. Nevertheless, it's important to note that the solution to Eq.(3) does not necessarily preserve the fixed sparsity pattern of $S'$ that is required to match $S$ in Eq.(2). To address this issue, we propose to incorporate the binary mask $P$ into the optimization process to ensure that the sparsity pattern of $S'$ is fixed as $S$. This is achieved by rewriting Eq.(2) as follows:

$$W = \underbrace{\overbrace{(W \odot P}^{S} - Q \odot P)}_{\text{sparse part } S'} + \underbrace{(W \odot (1 - P) + Q \odot P)}_{\text{low-rank part } L}. \quad (4)$$

Where $Q$ is a learnable matrix in the same shape as $W$.

*Baseline* 2 (PCP with mask). By directly substituting the decomposition from Eq.(4) into the optimization problem presented in Eq.(3) yields:

$$\min_{Q} \|W \odot (1 - P) + Q \odot P\|_* + \lambda \|W \odot P - Q \odot P\|_1,$$
$$\text{and } L = W \odot (1 - P) + Q \odot P. \quad (5)$$

This equation represents a constrained optimization problem where we seek to find the optimal $Q$. The binary mask $P$ plays a crucial role in maintaining the desired sparsity pattern. To solve this optimization problem, we can employ iterative methods such as gradient descent or its variants. Following Candès et al. (2011), we set $\lambda = 1/\sqrt{\max(m, n)}$. However, the nuclear norm minimization approach has a limitation: it applies equal shrinkage to all rank components (Zha et al., 2019). This uniform treatment may not be optimal for our objective in Eq.(2), where we aim to approximate the residual matrix $W - S$ using a low-rank matrix $L_k$ with a rank lower than that of $L$.

To address this limitation, we propose a forward-only method that prioritizes the preservation of low-rank components associated with larger singular values, while allowing for a more aggressive reduction of components with smaller singular values. This approach offers greater flexibility and precision in low-rank approximation compared to the PCP baseline, aligning more closely with our goal of achieving an efficient low-rank approximation. We begin by setting $S^{(0)} = W \odot P$, and then iteratively refine $S^{(t+1)}$ using the following update rule:

$$S^{(t+1)}$$
$$= S^{(t)} + P \odot \left\{ \sum_{i=r^{(t)}+1}^{r} \sigma_i \left( L^{(t)} \right) u_i \left( L^{(t)} \right) v_i^\top \left( L^{(t)} \right) \right\} \quad (6)$$
$$= S^{(t)} + P \odot \left\{ U_{r^{(t)}:} \left( L^{(t)} \right) \Sigma_{r^{(t)}:} \left( L^{(t)} \right) V_{r^{(t)}:}^\top \left( L^{(t)} \right) \right\}. \quad (7)$$

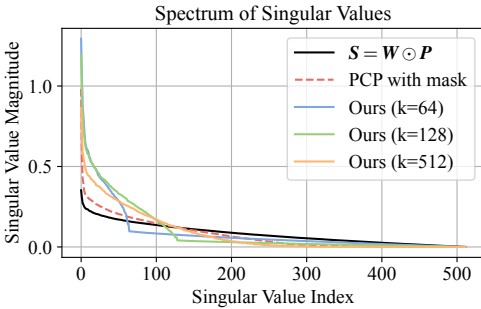

(a) Singular value distribution of $L$

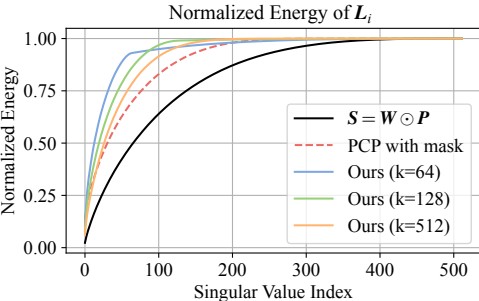

(b) Cumulative energy retention in $L_k$

Figure 2: Analysis of the residual matrix $L = W - S'$ and its low-rank approximation $L_k$ of zero-shot SVD, PCP baseline ($T = 5000$), and our method with varying $k$ $(64, 128, 512)$ and $T = 50$. (a) Singular value spectrum of $L$. (b) Proportion of total energy captured by the top $k$ singular values, calculated as $\sum_{j=1}^{i} \sigma_j^2(L_k)/\sum_{j=1}^{r} \sigma_j^2(L)$.

In this equation, $L^{(t)}$ represents the difference $W - S^{(t)}$, and $r^{(t)}$ is defined as $\lfloor 1 + \frac{k-1}{T-1}t \rfloor$, where $T$ denotes the total number of iterations, $k$ is the target rank of $L_k$, and $t$ ranges from 0 to $T - 1$. The linear schedule of $r^{(t)}$ serves several important purposes: (1) starting with a low rank forces the algorithm to first capture the most important singular components, which typically contain the majority of the energy, (2) each iteration then gradually incorporates more subtle details from higher singular components, and (3) this progressive approach helps maintain numerical stability and accelerates convergence compared to directly optimizing for the target rank $k$ from the beginning. We summarize the above algorithm in Algorithm 1. In each iteration, our algorithm focuses on eliminating the least significant components from the residual matrix $L^{(t)}$, which are associated with smaller singular values, while preserving those with larger singular values. This approach allows for a progressive refinement of the sparse matrix $S$. As we advance through the iterations, we gradually incorporate more subtle details from the original dense weight matrix $W$ into the updated sparse term $S$. For detailed guidelines on selecting the hyperparameters $k$ and $T$, we refer readers to Appendix D.

**Computational complexity analysis.** The primary com-

putational bottleneck in our algorithm is the SVD computation performed in each iteration. For a weight matrix $W \in \mathbb{R}^{m \times n}$ (assuming $m \geq n$), the time complexity of SVD is $O(mn^2)$. With $T$ iterations, the overall time complexity is $O(Tmn^2)$. However, our method remains significantly more efficient than comparable approaches, which typically require thousands of iterations. While the zero-shot SVD baseline is the fastest with only one SVD computation, it generally yields lower-quality results as shown in our experiments. Our method achieves substantially better performance than zero-shot SVD with relatively few iterations ($T = 50$). For a detailed analysis of the inference acceleration and memory usage of our method, please refer to Appendix E.

Incrementally increasing the rank $r^{(t)}$ from 2 to $k$ over $T$ iterations enables a more nuanced exploration of the weight space. Furthermore, the consistent application of the binary mask $P$ throughout this optimization process ensures the preservation of the desired sparsity pattern.

In Figure 2, we show the singular value spectrum of $L$ and the cumulative energy retention in $L_k$ for different methods using a $512 \times 512$ weight block extracted from a fine-tuned LLaMA-7B model. Subfigure 2a demonstrates that the proposed method with different target rank values $k$ (64, 128, and 512), consistently produces a more pronounced decay in singular values compared to both the zero-shot SVD and the PCP baseline. Subfigure 2b shows that the cumulative energy retention in $L$, calculated as $E(i) = \sum_{j=1}^{i} \sigma_j^2(L)/\sum_{j=1}^{i} \sigma_j^2(L)$ or $\|L_i\|_F^2/\|L\|_F^2$, increases more rapidly with truncated rank $i$ for the proposed method compared to the PCP baseline 2.

From Algorithm 1 and observations from Figure 2, we can summarize the technical contributions of our proposed method as follows: (1) *Iterative refinement with adaptive rank increase.* Our method progressively refines the sparse matrix $S$ while gradually increasing the rank $r^{(t)}$ from 2 to $k$ over $T$ iterations, prioritizing the most significant singular components of the residual matrix $L$ first and gradually incorporating more subtle details. (2) *Sparsity preservation.* Throughout the optimization process, our method strictly maintains the binary mask $P$, ensuring complete preservation of the desired sparsity pattern. This enables compatibility with both unstructured and structured pruning techniques for hardware-efficient implementations. This flexibility extends beyond the N:M structured sparsity patterns discussed in our experiments to other hardware-efficient patterns such as block sparsity (e.g., 4×4 blocks) and channel pruning since Theorem 4.1 guarantees that our method preserves any arbitrary sparsity pattern defined by the mask. (3) *Efficient information capture.* By focusing on preserving components with larger singular values first, our approach creates low-rank approximations that more efficiently capture the es-

sential information of the original weight matrix compared to alternative methods.

## 4.1. Theoretical Analysis

In this section, we delve into the theoretical underpinnings of our proposed method, offering a more rigorous analysis of its properties and performance. We present key theorems that elucidate the behavior of our algorithm, focusing on two critical aspects: sparsity preservation and convergence.

First, we demonstrate that our method maintains the desired sparsity pattern throughout the iterative process, ensuring that the final solution adheres to the specified binary mask. This property is crucial for applications where specific weight connections must remain zero.

**Theorem 4.1** (Sparsity Preservation, Proof. A.2.). *For all iterations $t$, the sparsity pattern of $S^{(t)}$ is preserved and matches the binary mask $P$, i.e., $S^{(t)} = S^{(t)} \odot P$ for all $t$.*

Second, we prove the convergence of our algorithm, showing that it approaches a well-defined solution as the number of iterations increases. This convergence guarantee provides theoretical justification for the stability and reliability of our method.

**Theorem 4.2** (Convergence). *For any weight matrix $W \in \mathbb{R}^{m \times n}$ and binary mask $P \in \{0,1\}^{m \times n}$, the iterative weight update algorithm converges to a solution $(S^*, L^*)$ as $T \to \infty$, such that $W = S^* + L^*$, $S^* = S^* \odot P$. Moreover, $\lim_{T \to \infty} \|S^{(T)} - S^*\|_F = 0$ and $\lim_{T \to \infty} \|L^{(T)} - L^*\|_F = 0$.*

**Theorem 4.3** (Asymptotic Convergence). *There exists a time step $T_0$ such that for all $t > T_0$, the Frobenius norm of the error decreases monotonically, i.e., for some*

$$\left\| W - \left( S^{(t+1)} + L_k^{(t+1)} \right) \right\|_F \leq \left\| W - \left( S^{(t)} + L_k^{(t)} \right) \right\|_F.$$
(8)

A corollary of Theorem. 4.3 is that if we fix $r^{(t)} = k$ for all $t$, the Frobenius norm of the error decreases monotonically. In this case, the term $\|P \odot L_{r^{(t)};k}^{(t)}\|_F^2$ vanishes, and Eq.(44) becomes $\|E^{(t+1)}\|_F^2 \leq \|E^{(t)}\|_F^2 - \|P \odot E^{(t)}\|_F^2$. Formally, we have the following corollary:

**Corollary 4.4** (Monotonic Improvement). *If we choose to fix $r^{(t)} = k$ for all $t$, the Frobenius norm of the error decreases monotonically, i.e.,*

$$\left\| E^{(t+1)} \right\|_F - \left\| E^{(t)} \right\|_F \leq - \left\| P \odot E^{(t)} \right\|_F \leq 0, \quad (9)$$

*where the equality holds if and only if $E^{(t)} = 0$.*

However, our empirical observations suggest that fixing $r^{(t)} = k$ throughout the process is not optimal in practice, and gradually increasing $r^{(t)}$ as the iterations progress leads to faster convergence.

**Theorem 4.5** (Error Bound). *At each iteration $t$, the Frobenius norm of the difference between the original weight matrix $W$ and its approximation $S^{(t)} + L_k^{(t)}$ is bounded by:*

$$\left\| W - \left( S^{(t)} + L_k^{(t)} \right) \right\|_F = \sqrt{ \sum_{i=k+1}^{r} \sigma_i^2 \left( L^{(t)} \right) } \qquad (10)$$

$$\leq \sqrt{(r-k)} \sigma_{k+1} \left( L^{(t)} \right), \quad (11)$$

*where $\sigma_i \left( L^{(t)} \right)$ are the singular values of $L^{(t)} = W - S^{(t)}$, and $r = rank \left( L^{(t)} \right)$.*

For a more comprehensive treatment, including detailed proofs of these theorems and additional supporting lemmas, we direct the reader to Appendix A. This appendix contains the full mathematical derivations and supplementary results that underpin our theoretical analysis.

# 5. Experiment

## 5.1. Experimental Setup

We conducted our experiments using LLaMA models, evaluating their performance on the WikiText-2 (Merity et al., 2016) and standard benchmarks including TruthfulQA (Lin et al., 2021), GSM8K (Cobbe et al., 2021), ARC-C (Clark et al., 2018) and MMLU (Hendrycks et al., 2020). We report perplexity on WikiText-2, where lower values indicate better performance. We evaluate various sparsity levels, including unstructured sparsity and structured N:M sparsity.

It is important to note that our proposed iterative refinement method is entirely data-free and does not require calibration data, as shown in Algorithm 1. We consistently use $T = 50$ across all experiments, which is sufficient for achieving most of the potential error reduction while maintaining computational efficiency. When implementing Wanda pruning (Sun et al., 2023) and our method combined with Wanda (Wanda + Ours), we use 128 sequences from the 'allenai/c4' dataset as calibration data. For evaluation, we use 128 sequences from WikiText-2 dataset for perplexity evaluation.

## 5.2. Effectiveness of Low-Rank Refinement

**Low-rank refinement advantage.** First, we aim to examine the benefits of low-rank refinement on WikiText-2. In Figure 3, we show the perplexity evaluation for various sparsity levels using magnitude pruning and low-rank refinement methods, with the target rank $k$ set to 128. In subfigure 3a, we compare the perplexity of sparsity-only pruning and zero-shot SVD refinement to highlight the low-rank refinement advantage. By incorporating a low-rank structure into the pruned weights, the parameter count increases. The x-axis represents the parameter reduction relative to the dense model instead of the sparsity level, and we use a dashed

Table 1: Comparison of WikiText validation perplexity (↓ is better) across various sparsity levels on LLaMA-7B. All methods and sparsity levels use a target rank of $k = 128$ (4.9% more parameters).

| METHOD | SPARSITY LEVEL (LLaMA-7B) | | | | |
| --- | --- | --- | --- | --- | --- |
| | **50%** | **60%** | **70%** | **4:8** | **2:4** |
| Dense | | | 5.68 | | |
| *Magnitude* | 17.29 (0%) | 152.36 (0%) | 48427.85 (0%) | 16.83 (0%) | 42.53 (0%) |
| w/ Zero-shot SVD | 8.06 (-53.4%) | 13.59 (-91.1%) | 283.74 (-99.4%) | 9.29 (-44.8%) | 12.72 (-70.0%) |
| w/ PCP with mask | 8.70 (-49.7%) | 16.67 (-89.1%) | 727.54 (-98.5%) | 10.60 (-37.0%) | 16.62 (-60.9%) |
| w/ **Ours** | **7.97 (-53.9%)** | **12.14 (-92.0%)** | **200.09 (-99.6%)** | **8.86 (-47.4%)** | **10.74 (-74.7%)** |
| *Wanda* | 7.26 (0%) | 10.69 (0%) | 84.69 (0%) | 8.57 (0%) | 11.53 (0%) |
| w/ Zero-shot SVD | 7.09 (-2.3%) | 9.60 (-10.2%) | 35.65 (-57.9%) | 8.14 (-5.0%) | 10.48 (-9.1%) |
| w/ PCP with mask | 7.28 (+0.3%) | 10.19 (-4.7%) | 47.11 (-44.4%) | 8.63 (-0.7%) | 11.22 (-2.7%) |
| w/ **Ours** | **6.92 (-4.7%)** | **8.97 (-16.1%)** | **32.90 (-61.2%)** | **7.74 (-9.7%)** | **9.18 (-20.4%)** |

Table 2: Comparison of WikiText validation perplexity (↓ is better) across various sparsity levels on LLaMA-13B. All methods and sparsity levels use a target rank of $k = 128$ (3.8% more parameters).

| METHOD | SPARSITY LEVEL (LLaMA-13B) | | | | |
| --- | --- | --- | --- | --- | --- |
| | **50%** | **60%** | **70%** | **4:8** | **2:4** |
| Dense | | | 4.57 | | |
| *Magnitude* | 5.98 (0%) | 9.91 (0%) | 408.75 (0%) | 6.76 (0%) | 8.32 (0%) |
| w/ Zero-shot SVD | 5.73 (-4.2%) | **8.83 (-10.9%)** | 163.96 (-59.9%) | 6.58 (-2.7%) | 8.86 (-6.5%) |
| w/ **Ours** | **5.65 (-5.5%)** | **8.83 (-10.9%)** | **99.27 (-75.7%)** | **6.40 (-5.3%)** | **7.76 (-6.7%)** |

black arrow to indicate the correspondence between the pruned model and the model with refinement of the same sparsity level. The results indicate that we gain 8.6% parameter reduction at approximately 50% sparsity, highlighting the superiority of low-rank refinement compared to sparsity-only pruning techniques. In subfigure 3b, we compare our method with other baseline methods. Our method consistently outperforms the baseline methods, with the benefits becoming more pronounced at higher sparsity levels. It is evident that PCP with mask under-performs compared to zero-shot SVD and our iterative weight update method, although it still surpasses sparsity-only pruning. This can be attributed to PCP's uniform shrinkage of all singular values, including smaller ones, and our choice of a very low target rank $k = 128$. To better understand the performance of PCP, we visualize the singular value spectrum of PCP in Figure 4a, Appendix B. PCP yields larger singular values for indices ranging from $10^2$ to nearly $10^3$, demonstrating its less discriminative approach to value reduction.

**Comparison across different sparsity types and levels.** To delve deeper into the effectiveness of low-rank refinement across varying sparsity types and levels, we offer a comprehensive comparison of WikiText-2 validation perplexity using LLaMA-7B and 13B. The results are presented in Tables 1 and 2 respectively. Where we show the experimen-

tal results for different sparsity levels, ranging from 50% to 70%, as well as structured sparsity patterns like 4:8 and 2:4. For each sparsity level, we compare the performance of low-rank refinement incorporated with magnitude pruning and Wanda pruning (Sun et al., 2023). At 50% sparsity, our method reduces perplexity by 53.9% compared to sparse-only magnitude pruning. This improvement becomes more significant at higher sparsity levels, reaching a 92.0% reduction at 60% sparsity and a 99.6% reduction at 70% sparsity. Our proposed method consistently achieves lower perplexity values across all sparsity levels. This detailed comparison highlights the effectiveness of our method in maintaining low perplexity even at higher sparsity levels, demonstrating its robustness and superiority over other baseline methods.

**Benchmark results.** In Table 3, we evaluate the performance of our method on several benchmark datasets. We compare the performance of our method with that of dense models, sparsity-only pruned models, and low-rank refined using zero-shot SVD. Our method consistently outperforms both magnitude pruning and zero-shot SVD across most tasks and model sizes. For the 7B model, it achieves a 13.2% reduction in average performance compared to 22.1% for magnitude pruning and 18.5% for zero-shot SVD.

We further validate our approach on the more recent

Table 3: Performance comparison on LLaMA models across several benchmark datasets ($k = 128$).

| | METHOD | TruthfulQA | GSM8K | ARC-c | MMLU | AVG. |
|---|---|---|---|---|---|---|
| | Dense | 34.1 | 10.3 | 44.7 | 32.1 | 30.3 (0%) |
| 7B | *Magnitude 50%* | **35.3** | 1.0 | 33.5 | 24.6 | 23.6 (-22.1%) |
| | w/ Zero-shot SVD | 34.3 | 1.5 | 36.9 | **26.0** | 24.7 (-18.5%) |
| | w/ **Ours** | 34.2 | **3.4** | **41.5** | **26.0** | **26.3 (-13.2%)** |
| | *SparseGPT 2:4* | **36.5** | 2.0 | 33.1 | 25.4 | 24.3 (-19.8%) |
| | w/ **Ours** | 33.8 | **2.7** | **36.1** | **29.1** | 25.4 (-16.2%) |
| | Dense | 36.9 | 23.4 | 49.1 | 52.1 | 40.4 (0%) |
| 13B | *Magnitude 2:4* | **38.4** | 1.7 | 34.9 | 27.8 | 25.7 (-36.4%) |
| | w/ Zero-shot SVD | 37.6 | 1.8 | 32.2 | 27.0 | 24.7 (-38.9%) |
| | w/ **Ours** | 36.9 | **9.4** | **36.7** | **41.9** | **31.2 (-22.8%)** |

LLaMA-3.1-8B model, which has enhanced reasoning and mathematical capabilities compared to LLaMA-2. Detailed results are presented in Table 4 in Appendix C. On this frontier model, our method significantly improves performance over magnitude pruning across most benchmarks. Notably, on GSM8K, which tests mathematical reasoning, the dense model achieves 49.8% accuracy while magnitude unstructured pruning at 50% sparsity reduces this to just 1.3%. Our low-rank refinement method improves this to 6.5%, showing substantial recovery of the mathematical reasoning capabilities. Similar improvements are observed across other benchmarks, with our method recovering more than 10 percentage points on HellaSwag, WinoGrande, ARC-e, ARC-c, and MMLU compared to magnitude pruning alone.

## 5.3. Iterative Weight Update Analysis

In this section, we examine the performance of our proposed Algorithm 1 to empirically confirm the theoretical properties outlined in Section 4, specifically regarding convergence properties, error reduction, and the lower bound of rank($L$).

**Convergence and error analysis.** In Figure 4, we visualize several key properties of the residual matrix $L = W - S'$ and its low-rank approximation $L_k$, we use a log scale for the $x$-axis to better visualize the decay of singular values at low ranks. Throughout the analysis, we use magnitude pruning with a sparsity level of 50%. (a) *Singular Value Spectrum*. We first visualize the singular value spectrum of $L$ for different methods and hyperparameter configurations for our method. It is clear that the iterative weight update method exhibits a more pronounced decay in singular values compared to both zero-shot SVD and the PCP baseline. (b) *Energy Retention*. Here we show the cumulative energy retention of $L$, calculated as $\sum_{j=1}^{i} \sigma_j^2(L_k) / \sum_{j=1}^{r} \sigma_j^2(L) = \|L_i\|_F^2 / \|L\|_F^2$. Across various target rank values, our method captures a higher proportion of energy within the top $k$ singular values, suggesting

that our low-rank approximations preserve more information from the original matrix, thereby enhancing performance. (c) *Error Analysis*. This subfigure presents the Frobenius norm of the error $E_i = W - (S^{(t)} + L_i^{(t)})$, noting that $E_i = L - L_i$ as well. It is observed that at the desired target rank, our method achieves a lower Frobenius norm error than both zero-shot SVD and the PCP baseline. (d) *Convergence Analysis*. We illustrate how the approximation error between the dense matrix $W$ and the combined approximation $S^{(t)} + L_k^{(t)}$ diminishes over iterations, as predicted by Theorem 4.2 and Theorem 4.3. Specifically, it shows the Frobenius norm $\|E_i^{(t)}\|_F = \|L^{(t)} - L_i^{(t)}\|_F$ at various iterations for $k = 512$ and $T = 50$.

**Investigating the lower bound of rank($L$)** Following previous analysis, an intriguing question arises: *Is it possible to fully compress the information contained in $W \odot (1 - P)$ into $S'$? In other words, does the rank of $L$ have a lower bound?* To address this, we conduct an empirical investigation by applying our method to various target ranks $k$, using both synthetic matrices and real-world weight matrices extracted from the LLaMA-7B model. We first construct a series of synthetic matrices $W$ of known rank $r$ by multiplying two Gaussian random matrices: $U \in \mathbb{R}^{m \times r}$ and $V \in \mathbb{R}^{n \times r}$, such that $W = UV^T$. Here, $m = n = 512$. We subsequently apply our method along with zero-shot SVD to $W$, varying the rank $r$ and sparsity level $p$, and visualize the spectrum of $L = W - S'$ in Figure 5

In both subfigures in Figure 5, Appendix B, the solid lines represent the zero-shot SVD baseline method ($L = W \odot (1 - P)$), while the dashed lines represent the proposed method. The proposed method consistently shows a steeper decay in singular values compared to the zero-shot SVD method, indicating better compression of information. In subfigure 5a, we observe that for diverse ranks of $W$ and a constant sparsity level $p = 0.5$, the rank of $L$ remains invariant. Conversely, subfigure 5b shows that fixing the

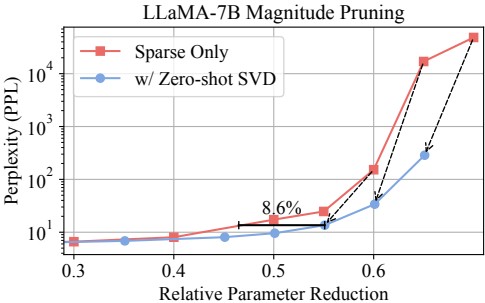

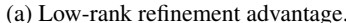

(a) Low-rank refinement advantage.

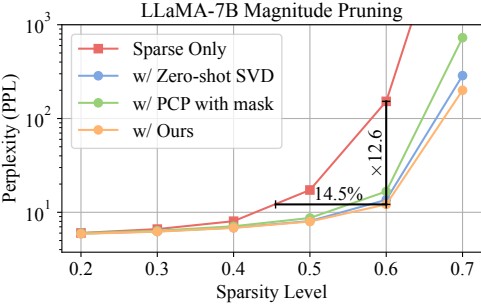

(b) Method comparison.

Figure 3: Perplexity evaluation for different sparsity levels and methods, $k = 128$ for low-rank refinement. (a) Low-rank refinement advantage over sparsity-only pruning. To ensure fair comparison between methods, in this plot, we report results based on total parameter count rather than just sparsity level. (b) Comparison of our proposed iterative weight update method with other baseline methods.

rank of $W$ at $r = 512$ and varying the sparsity level $p$ from 0.3 to 0.7, reveals a positive correlation between the rank of $L$ and increasing sparsity. More specifically, the rank of $L$ is around $p \times \min(m, n)$. Empirically, we come to the conclusion that the rank of $L$ is independent of $\text{rank}(W)$ but dependent on the sparsity level $p$. This is consistent with the intuitive explanation that as the sparsity level increases, the pruned matrix contains more information, and thus the rank of $L$ is higher. Similar observations can be made in the LLaMA-7B model, as shown in Figure 2.

## 6. Conclusion

In this work, we present an iterative weight update algorithm for low-rank refinement of sparse-pruned models. Our approach aims to bridge the performance gap between dense and pruned sparse models in large language models. The proposed method offers a computationally efficient solution that does not rely on extensive datasets or high-performing teacher models, making it a practical choice for improving sparse model performance. A notable advantage of our method is its sparsity-preserving property, which allows for the concurrent update of the sparse matrix while maintaining

its sparsity pattern and incorporating a low-rank component. This approach effectively recovers crucial information lost during pruning, leading to performance recovery, especially at high sparsity ratios.

These experimental results highlight the potential of combining low-rank and sparsity in LLMs. Future work could explore methods for automatically determining the optimal rank for the low-rank component based on the specific characteristics of each layer or the overall model architecture.

## Acknowledgements

This work is supported by STI 2030—Major Projects (No. 2021ZD0201405), Shenzhen Basic Research Project (Natural Science Foundation) Basic Research Key Project (NO. JCYJ20241202124430041), National Natural Science Foundation of China (No. 62441619, 62276195 and U23A20318), Open Foundation of Key Laboratory of Cyberspace Security, Ministry of Education of China (No.KLCS20240208).

## Impact Statement

Our study focuses on enhancing model performance through auxiliary-task learning, the broader impacts could be considered from various aspects as follows: The proposed method could lead to more efficient machine learning models that require less data to achieve high performance. This could benefit areas where data collection is expensive or difficult. Better-performing models could also be particularly useful in high-stakes domains such as healthcare (e.g., improved diagnostics through better image analysis) or autonomous systems (e.g., safer navigation through enhanced scene understanding). However, if the auxiliary tasks introduce biased data, the merged model might amplify these biases, leading to unfair or discriminatory outcomes in applications like recruitment systems or criminal justice. Future work could investigate this further.

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

## Targeted Low-rank Refinement: Enhancing Sparse Language Models with Precision

The appendix is divided into several sections, each giving extra information and details.

## A. Theoretical Analysis

In this section, we provide the proofs for the theorems presented in the main text and additional lemmas.

### A.1. Additional Lemmas

In this subsection, we present additional lemmas that are useful for the proofs of the theorems in the main text.

**Lemma A.1** (Mask Norm Inequality I). *For any matrix $\boldsymbol{M} \in \mathbb{R}^{m \times n}$ and binary mask $\boldsymbol{P} \in \{0,1\}^{m \times n}$, the following inequality holds:*

$$\|\boldsymbol{P} \odot \boldsymbol{M}\|_F \leq \|\boldsymbol{M}\|_F \tag{12}$$

*Proof of Lemma A.1.* Let $m_{ij}$ and $p_{ij}$ be the elements of $\boldsymbol{M}$ and $\boldsymbol{P}$ respectively. By definition of the Frobenius norm and element-wise multiplication:

$$\|\boldsymbol{P} \odot \boldsymbol{M}\|_F^2 = \sum_{i=1}^{m} \sum_{j=1}^{n} (p_{ij} m_{ij})^2 = \sum_{i=1}^{m} \sum_{j=1}^{n} p_{ij}^2 m_{ij}^2 \tag{13}$$

Since $\boldsymbol{P}$ is a binary mask, $p_{ij} \in \{0,1\}$, which means $p_{ij}^2 = p_{ij}$. Therefore:

$$\|\boldsymbol{P} \odot \boldsymbol{M}\|_F^2 = \sum_{i=1}^{m} \sum_{j=1}^{n} p_{ij} m_{ij}^2 \leq \sum_{i=1}^{m} \sum_{j=1}^{n} m_{ij}^2 = \|\boldsymbol{M}\|_F^2 \tag{14}$$

Taking the square root of both sides preserves the inequality:

$$\|\boldsymbol{P} \odot \boldsymbol{M}\|_F \leq \|\boldsymbol{M}\|_F \tag{15}$$

$\square$

**Lemma A.2** (Mask Norm Inequality II). *For any matrix $\boldsymbol{M} \in \mathbb{R}^{m \times n}$ and binary mask $\boldsymbol{P} \in \{0,1\}^{m \times n}$, the following inequality holds:*

$$\langle \boldsymbol{M}, \boldsymbol{P} \odot \boldsymbol{M} \rangle_F \leq \langle \boldsymbol{M}, \boldsymbol{M} \rangle_F \tag{16}$$

*Proof of Lemma A.2.* Let $m_{ij}$ and $p_{ij}$ be the elements of $\boldsymbol{M}$ and $\boldsymbol{P}$ respectively. By definition of the Frobenius inner product and element-wise multiplication:

$$\langle \boldsymbol{M}, \boldsymbol{P} \odot \boldsymbol{M} \rangle_F = \sum_{i=1}^{m} \sum_{j=1}^{n} m_{ij}(p_{ij} m_{ij}) = \sum_{i=1}^{m} \sum_{j=1}^{n} p_{ij} m_{ij}^2 \leq \sum_{i=1}^{m} \sum_{j=1}^{n} m_{ij}^2 = \langle \boldsymbol{M}, \boldsymbol{M} \rangle_F \tag{17}$$

$\square$

**Lemma A.3** (Mask Norm Equality I). *For any matrix $\boldsymbol{M} \in \mathbb{R}^{m \times n}$ and binary mask $\boldsymbol{P} \in \{0,1\}^{m \times n}$, the following equality holds:*

$$\langle \boldsymbol{M}, \boldsymbol{P} \odot \boldsymbol{M} \rangle_F = \|\boldsymbol{P} \odot \boldsymbol{M}\|_F^2 \tag{18}$$

*Proof of Lemma A.3.* Because $\boldsymbol{P}$ is a binary mask, we have $p_{ij}^2 = p_{ij}$. Therefore, we can rewrite the Frobenius inner product as:

$$\langle \boldsymbol{M}, \boldsymbol{P} \odot \boldsymbol{M} \rangle_F = \sum_{i=1}^m \sum_{j=1}^n m_{ij}(p_{ij}m_{ij}) = \sum_{i=1}^m \sum_{j=1}^n p_{ij}^2 m_{ij}^2 = \|\boldsymbol{P} \odot \boldsymbol{M}\|_F^2 \tag{19}$$

$\square$

**Lemma A.4** (Mask Norm Equality II). *For any matrix $\boldsymbol{A}, \boldsymbol{B} \in \mathbb{R}^{m \times n}$ and binary mask $\boldsymbol{P} \in \{0,1\}^{m \times n}$, the following equality holds:*

$$\langle \boldsymbol{P} \odot \boldsymbol{A}, \boldsymbol{P} \odot \boldsymbol{B} \rangle_F = \langle \boldsymbol{A}, \boldsymbol{P} \odot \boldsymbol{B} \rangle_F = \langle \boldsymbol{P} \odot \boldsymbol{A}, \boldsymbol{B} \rangle_F \tag{20}$$

*Proof of Lemma A.4.* By definitions of Frobenius inner product and element-wise multiplication, we have:

$$\langle \boldsymbol{P} \odot \boldsymbol{A}, \boldsymbol{P} \odot \boldsymbol{B} \rangle_F = \sum_{i=1}^m \sum_{j=1}^n (p_{ij}a_{ij})(p_{ij}b_{ij}) = \sum_{i=1}^m \sum_{j=1}^n p_{ij}^2 a_{ij} b_{ij} \tag{21}$$

$$\langle \boldsymbol{A}, \boldsymbol{P} \odot \boldsymbol{B} \rangle_F = \sum_{i=1}^m \sum_{j=1}^n a_{ij}(p_{ij}b_{ij}) = \sum_{i=1}^m \sum_{j=1}^n p_{ij} a_{ij} b_{ij} \tag{22}$$

$$\langle \boldsymbol{P} \odot \boldsymbol{A}, \boldsymbol{B} \rangle_F = \sum_{i=1}^m \sum_{j=1}^n (p_{ij}a_{ij})b_{ij} = \sum_{i=1}^m \sum_{j=1}^n p_{ij} a_{ij} b_{ij} \tag{23}$$

For binary mask, $p_{ij}^2 = p_{ij}$, so the three expressions are equivalent. $\square$

### A.2. Proofs of Theorems

*Proof of Theorem 4.1.* We prove this by induction. For $t = 0$, $\boldsymbol{S}^{(0)} = \boldsymbol{W} \odot \boldsymbol{P}$, so the property holds. Assume the property holds for iteration $t$, i.e., $\boldsymbol{S}^{(t)} = \boldsymbol{S}^{(t)} \odot \boldsymbol{P}$. We need to prove that it holds for iteration $t + 1$, i.e., $\boldsymbol{S}^{(t+1)} = \boldsymbol{S}^{(t+1)} \odot \boldsymbol{P}$. Therefore, we have:

$$\begin{aligned}
\boldsymbol{S}^{(t+1)} &= \boldsymbol{S}^{(t)} + \boldsymbol{P} \odot \left\{ \boldsymbol{U}_{r^{(t)}:}\left(\boldsymbol{L}^{(t)}\right) \boldsymbol{\Sigma}_{r^{(t)}:}\left(\boldsymbol{L}^{(t)}\right) \boldsymbol{V}_{r^{(t)}:}^{\top}\left(\boldsymbol{L}^{(t)}\right) \right\} \\
&= (\boldsymbol{S}^{(t)} \odot \boldsymbol{P}) + \boldsymbol{P} \odot (\text{SVD terms}) \quad \text{(applying induction hypothesis)} \\
&= \boldsymbol{P} \odot (\boldsymbol{S}^{(t)} + \text{SVD terms}) = \boldsymbol{P} \odot \boldsymbol{S}^{(t+1)}
\end{aligned}$$

Thus, we have shown that if the property holds for $t$, it also holds for $t + 1$. Combined with the base case, this completes the induction proof, showing that $\boldsymbol{S}^{(t)} = \boldsymbol{S}^{(t)} \odot \boldsymbol{P}$ for all $t$. $\square$

*Proof of Theorem 4.2.* Let $\boldsymbol{S}^*$ and $\boldsymbol{L}^*$ be the limits of $\boldsymbol{S}^{(T)}$ and $\boldsymbol{L}^{(T)}$ as $T \to \infty$, respectively. We will show that these limits exist and satisfy the stated properties. First, note that for any $t$, $\boldsymbol{S}^{(t)} = \boldsymbol{S}^{(t)} \odot \boldsymbol{P}$ by construction of the algorithm. Let $\epsilon_t = \|\boldsymbol{S}^{(t+1)} - \boldsymbol{S}^{(t)}\|_F$. We need to show that $\lim_{T \to \infty} \epsilon_{T-1} = 0$. At each iteration $t$, we have:

$$\boldsymbol{S}^{(t+1)} = \boldsymbol{S}^{(t)} + \boldsymbol{P} \odot \left\{ \boldsymbol{U}_{r^{(t)}:}\left(\boldsymbol{L}^{(t)}\right) \boldsymbol{\Sigma}_{r^{(t)}:}\left(\boldsymbol{L}^{(t)}\right) \boldsymbol{V}_{r^{(t)}:}^{\top}\left(\boldsymbol{L}^{(t)}\right) \right\} \tag{24}$$

$$= \boldsymbol{S}^{(t)} + \boldsymbol{P} \odot \left(\boldsymbol{W} - \boldsymbol{S}^{(t)}\right)_{r^{(t)}:} \tag{25}$$

where $(\cdot)_{r^{(t)}:}$ denotes the truncated SVD reconstruction from $r^{(t)}$ onwards. From the properties of SVD, we can express the difference between the truncated SVDs as:

$$\left(\boldsymbol{W} - \boldsymbol{S}^{(t)}\right)_{r^{(t)}:} - \left(\boldsymbol{W} - \boldsymbol{S}^{(t)}\right)_{k:} = \sum_{i=r^{(t)}+1}^{k} \sigma_i^{(t)} \boldsymbol{u}_i^{(t)} \boldsymbol{v}_i^{(t)\top}, \tag{26}$$

where $\sigma_i^{(t)}$, $\boldsymbol{u}_i^{(t)}$, and $\boldsymbol{v}_i^{(t)}$ are the $i$-th singular value and corresponding left and right singular vectors of $\boldsymbol{W} - \boldsymbol{S}^{(t)}$. The Frobenius norm of this difference is:

$$\left\| \sum_{i=r^{(t)}+1}^{k} \sigma_i^{(t)} \boldsymbol{u}_i^{(t)} \boldsymbol{v}_i^{(t)\top} \right\|_F^2 = \sum_{i=r^{(t)}+1}^{k} \sigma_i^{(t)2}. \tag{27}$$

While we cannot directly guarantee that this sum approaches zero as $t$ increases, we can bound it as:

$$\sum_{i=r^{(t)}+1}^{k} \sigma_i^{(t)2} \leq \left(k - r^{(t)}\right) \left(\sigma_{r^{(t)}+1}^{(t)}\right)^2. \tag{28}$$

As $t \to T - 1$, $r^{(t)} \to k$, so $k - r^{(t)} \to 0$. Thus, for any $\epsilon > 0$, there exists a $T_0$ such that for all $t > T_0$, we have $\left(k - r^{(t)}\right) \left(\sigma_{r^{(t)}+1}^{(t)}\right)^2 < \epsilon^2$. Now, considering the effect of the binary mask $\boldsymbol{P}$, according to Lemma A.1, we have:

$$\left\| \boldsymbol{P} \odot \left( \left(\boldsymbol{W} - \boldsymbol{S}^{(t)}\right)_{r^{(t)}:} - \left(\boldsymbol{W} - \boldsymbol{S}^{(t)}\right)_{k:} \right) \right\|_F \leq \left\| \left(\boldsymbol{W} - \boldsymbol{S}^{(t)}\right)_{r^{(t)}:} - \left(\boldsymbol{W} - \boldsymbol{S}^{(t)}\right)_{k:} \right\|_F < \epsilon \tag{29}$$

Setting $\delta = \epsilon$, we have shown that for any $\delta > 0$, there exists a $T_0$ such that for all $t > T_0$:

$$\left\| \boldsymbol{P} \odot \left(\boldsymbol{W} - \boldsymbol{S}^{(t)}\right)_{r^{(t)}:} - \boldsymbol{P} \odot \left(\boldsymbol{W} - \boldsymbol{S}^{(t)}\right)_{k:} \right\|_F < \delta \tag{30}$$

This implies that for $t > T_0$:

$$\epsilon_t = \|\boldsymbol{S}^{(t+1)} - \boldsymbol{S}^{(t)}\|_F < \delta \tag{31}$$

Since $\delta$ can be arbitrarily small, we conclude that $\lim_{T \to \infty} \epsilon_{T-1} = 0$. This shows that the sequence $\{\boldsymbol{S}^{(T)}\}$ is Cauchy and therefore converges to some limit $\boldsymbol{S}^*$. Since $\boldsymbol{W} = \boldsymbol{S}^{(t)} + \boldsymbol{L}^{(t)}$ for all $t$, and $\boldsymbol{S}^{(T)}$ converges to $\boldsymbol{S}^*$, it follows that $\boldsymbol{L}^{(T)}$ must converge to $\boldsymbol{L}^* = \boldsymbol{W} - \boldsymbol{S}^*$. Finally, since $\boldsymbol{S}^{(t)} = \boldsymbol{S}^{(t)} \odot \boldsymbol{P}$ for all $t$, we have $\boldsymbol{S}^* = \boldsymbol{S}^* \odot \boldsymbol{P}$. Thus, the algorithm converges to the solution $(\boldsymbol{S}^*, \boldsymbol{L}^*)$ as $T \to \infty$, satisfying all the stated properties. $\qquad \square$

*Proof of Theorem 4.3.* Let $\boldsymbol{E}^{(t)} = \boldsymbol{W} - \left(\boldsymbol{S}^{(t)} + \boldsymbol{L}_k^{(t)}\right) = \left(\boldsymbol{L}^{(t)}\right)_{k:}$ be the error at iteration $t$. From the update rule in Eq.(7), we have:

$$\boldsymbol{E}^{(t+1)} = \boldsymbol{W} - \left(\boldsymbol{S}^{(t+1)} + \boldsymbol{L}_k^{(t+1)}\right) \tag{32}$$

$$= \boldsymbol{W} - \left(\boldsymbol{S}^{(t)} + \boldsymbol{P} \odot \left(\boldsymbol{L}^{(t)}\right)_{r^{(t)}:} + \boldsymbol{L}_k^{(t+1)}\right) \tag{33}$$

$$= \boldsymbol{L}^{(t)} - \boldsymbol{P} \odot \left(\boldsymbol{L}^{(t)}\right)_{r^{(t)}:} - \boldsymbol{L}_k^{(t+1)} \tag{34}$$

Let's denote $\boldsymbol{A} = \boldsymbol{P} \odot \left(\boldsymbol{L}^{(t)}\right)_{r^{(t)}:}$. Then $\boldsymbol{E}^{(t+1)} = \left(\boldsymbol{L}^{(t)} - \boldsymbol{A}\right) - \boldsymbol{L}_k^{(t+1)}$. By construction, $\boldsymbol{L}_k^{(t+1)}$ is the best rank-k approximation of $\boldsymbol{L}^{(t+1)} = \boldsymbol{L}^{(t)} - \boldsymbol{A}$. Therefore:

$$\left\| \boldsymbol{E}^{(t+1)} \right\|_F = \left\| \left(\boldsymbol{L}^{(t)} - \boldsymbol{A}\right) - \boldsymbol{L}_k^{(t+1)} \right\|_F \leq \left\| \left(\boldsymbol{L}^{(t)} - \boldsymbol{A}\right) - \boldsymbol{L}_k^{(t)} \right\|_F \tag{35}$$

Now, $\boldsymbol{L}^{(t)} - \boldsymbol{L}_k^{(t)} = \boldsymbol{E}^{(t)}$, so we have:

$$\left\| \boldsymbol{E}^{(t+1)} \right\|_F^2 \leq \left\| \left(\boldsymbol{L}^{(t)} - \boldsymbol{A}\right) - \boldsymbol{L}_k^{(t)} \right\|_F^2 = \left\| \left(\boldsymbol{L}^{(t)} - \boldsymbol{L}_k^{(t)}\right) - \boldsymbol{A} \right\|_F^2 \tag{36}$$

$$= \left\| \boldsymbol{E}^{(t)} - \boldsymbol{A} \right\|_F^2 = \left\| \boldsymbol{E}^{(t)} \right\|_F^2 + \|\boldsymbol{A}\|_F^2 - 2 \left\langle \boldsymbol{E}^{(t)}, \boldsymbol{A} \right\rangle_F \tag{37}$$

According to Lemma A.2 and $\boldsymbol{A} = \boldsymbol{P} \odot \left( \boldsymbol{L}_{r^{(t)}:k}^{(t)} + \boldsymbol{E}^{(t)} \right)$, we have:

$$\left\langle \boldsymbol{E}^{(t)}, \boldsymbol{A} \right\rangle_F = \left\langle \boldsymbol{E}^{(t)}, \boldsymbol{P} \odot \left( \boldsymbol{L}_{r^{(t)}:k}^{(t)} + \boldsymbol{E}^{(t)} \right) \right\rangle_F \tag{38}$$

$$= \left\langle \boldsymbol{E}^{(t)}, \boldsymbol{P} \odot \boldsymbol{L}_{r^{(t)}:k}^{(t)} \right\rangle_F + \left\langle \boldsymbol{E}^{(t)}, \boldsymbol{P} \odot \boldsymbol{E}^{(t)} \right\rangle_F \tag{39}$$

$$= \left\langle \boldsymbol{E}^{(t)}, \boldsymbol{P} \odot \boldsymbol{L}_{r^{(t)}:k}^{(t)} \right\rangle_F + \left\| \boldsymbol{P} \odot \boldsymbol{E}^{(t)} \right\|_F^2 \tag{40}$$

The last equality follows from Lemma A.3. Substituting this back into our earlier inequality Eq.(37):

$$\left\| \boldsymbol{E}^{(t+1)} \right\|_F^2 \le \left\| \boldsymbol{E}^{(t)} \right\|_F^2 + \|\boldsymbol{A}\|_F^2 - 2 \left\langle \boldsymbol{E}^{(t)}, \boldsymbol{A} \right\rangle_F \tag{41}$$

$$\le \left\| \boldsymbol{E}^{(t)} \right\|_F^2 + \|\boldsymbol{A}\|_F^2 - 2 \left\langle \boldsymbol{E}^{(t)}, \boldsymbol{P} \odot \boldsymbol{L}_{r^{(t)}:k}^{(t)} \right\rangle_F - 2 \left\| \boldsymbol{P} \odot \boldsymbol{E}^{(t)} \right\|_F^2 \tag{42}$$

$$= \left\| \boldsymbol{E}^{(t)} \right\|_F^2 + \left\| \boldsymbol{P} \odot \boldsymbol{L}_{r^{(t)}:k}^{(t)} \right\|_F^2 + \left\| \boldsymbol{P} \odot \boldsymbol{E}^{(t)} \right\|_F^2 + 2 \overline{\left\langle \boldsymbol{P} \odot \boldsymbol{L}_{r^{(t)}:k}^{(t)}, \boldsymbol{P} \odot \boldsymbol{E}^{(t)} \right\rangle_F}$$
$$\overline{-2 \left\langle \boldsymbol{E}^{(t)}, \boldsymbol{P} \odot \boldsymbol{L}_{r^{(t)}:k}^{(t)} \right\rangle_F} - 2 \left\| \boldsymbol{P} \odot \boldsymbol{E}^{(t)} \right\|_F^2 \qquad \text{(by Lemma. A.4)} \tag{43}$$

$$= \left\| \boldsymbol{E}^{(t)} \right\|_F^2 + \left\| \boldsymbol{P} \odot \boldsymbol{L}_{r^{(t)}:k}^{(t)} \right\|_F^2 - \left\| \boldsymbol{P} \odot \boldsymbol{E}^{(t)} \right\|_F^2 \tag{44}$$

As $t \to T - 1$, $r^{(t)} \to k$, so $\left\| \boldsymbol{P} \odot \boldsymbol{L}_{r^{(t)}:k}^{(t)} \right\|_F^2 \to 0$. Therefore, there exists a time step $T_0$ such that for all $t > T_0$, $\left\| \boldsymbol{E}^{(t+1)} \right\|_F^2 - \left\| \boldsymbol{E}^{(t)} \right\|_F^2 \le - \left\| \boldsymbol{P} \odot \boldsymbol{E}^{(t)} \right\|_F^2 \le 0$. $\qquad\square$

*Proof of Theorem 4.5.* At iteration $t$, we have $\boldsymbol{W} = \boldsymbol{S}^{(t)} + \boldsymbol{L}^{(t)}$ by construction. Let $\boldsymbol{L}^{(t)} = \boldsymbol{U}\boldsymbol{\Sigma}\boldsymbol{V}^\top$ be the SVD of $\boldsymbol{L}^{(t)}$. Then $\boldsymbol{L}_k^{(t)} = \boldsymbol{U}_{:k}\boldsymbol{\Sigma}_{:k}\boldsymbol{V}_{:k}^\top$ is the best rank-k approximation of $\boldsymbol{L}^{(t)}$. The error can be expressed as:

$$\left\| \boldsymbol{W} - \left( \boldsymbol{S}^{(t)} + \boldsymbol{L}_k^{(t)} \right) \right\|_F = \left\| \left( \boldsymbol{S}^{(t)} + \boldsymbol{L}^{(t)} \right) - \left( \boldsymbol{S}^{(t)} + \boldsymbol{L}_k^{(t)} \right) \right\|_F \tag{45}$$

$$= \left\| \boldsymbol{L}^{(t)} - \boldsymbol{L}_k^{(t)} \right\|_F = \| \boldsymbol{U}_{k:}\boldsymbol{\Sigma}_{k:}\boldsymbol{V}_{k:}^\top \|_F \tag{46}$$

$$= \sqrt{\sum_{i=k+1}^{r} \sigma_i^2 \left( \boldsymbol{L}^{(t)} \right)} \le \sqrt{(r-k)}\sigma_{k+1} \left( \boldsymbol{L}^{(t)} \right) \tag{47}$$

$$\square$$

## B. Additional Experimental Results of Iterative Weight Update Analysis

In this section, we provide additional experimental results of iterative weight update analysis.

In Figure 4, we present a comprehensive analysis of the residual matrix $\boldsymbol{L} = \boldsymbol{W} - \boldsymbol{S}'$ and compare various methods for computing its low-rank approximation $\boldsymbol{L}_k$. The figure illustrates four key aspects: (a) the distribution of singular values of $\boldsymbol{L}$, (b) the cumulative energy retention across different singular values, calculated as $\sum_{i=1}^{k} \sigma_i^2(\boldsymbol{L}_k) / \sum_{i=1}^{r} \sigma_i^2(\boldsymbol{L})$ (c) the approximation error in terms of Frobenius norm, $\|\boldsymbol{L} - \boldsymbol{L}_i\|_F$, and (d) the convergence behavior across iterations.

In Figure 5, we analyze how the singular value spectrum of $\boldsymbol{L}$ varies under different experimental conditions using synthetic weight matrices $\boldsymbol{W}$. We compare our proposed method against the zero-shot SVD baseline ($\boldsymbol{L} = \boldsymbol{W} \odot (1 - \boldsymbol{P})$) in two scenarios: (a) varying the rank of the original weight matrix $\boldsymbol{W}$ (64, 128, 256, 512), and (b) different sparsity levels (0.3, 0.5, 0.7) achieved through magnitude pruning. This analysis helps us understand how the structural properties of the residual matrix are affected by these key parameters.

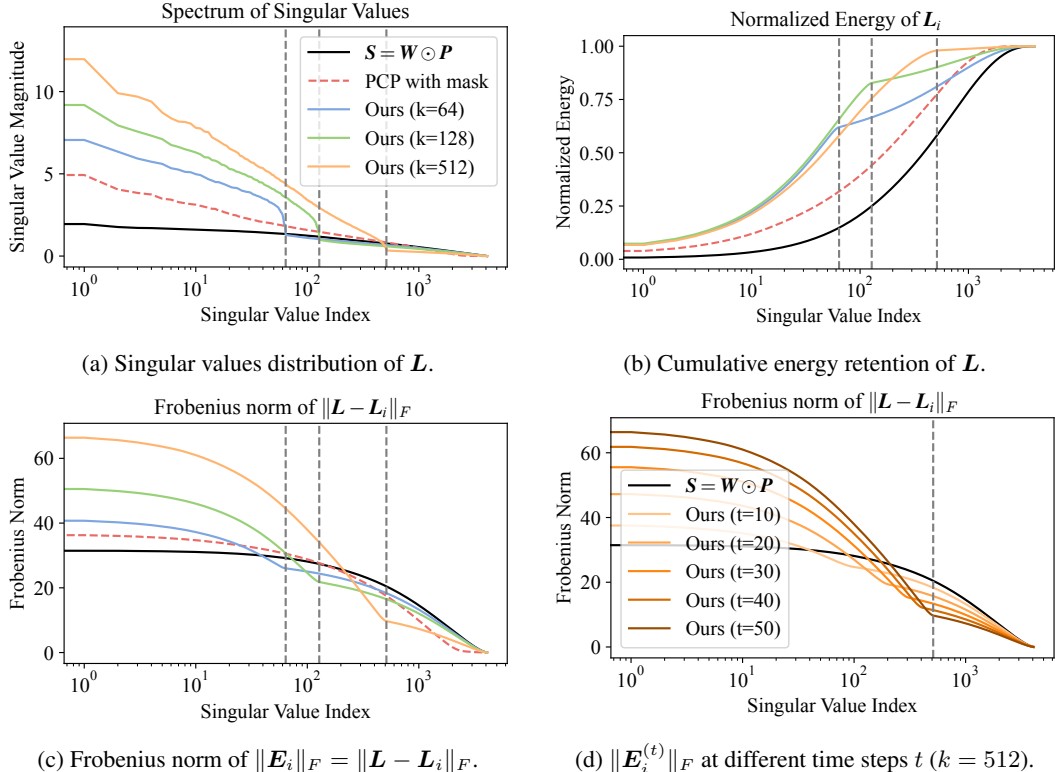

(a) Singular values distribution of $L$.

(b) Cumulative energy retention of $L$.

(c) Frobenius norm of $\|E_i\|_F = \|L - L_i\|_F$.

(d) $\|E_i^{(t)}\|_F$ at different time steps $t$ ($k = 512$).

Figure 4: Analysis of the residual matrix $L = W - S'$ and its low-rank approximation $L_k$ using different methods. Results are shown for zero-shot SVD, PCP baseline ($T = 100$), and our proposed method with varying $k$ (64, 128, 512) and $T = 50$. We show the $x$-axis in log scale and vertical dashed lines at $i = 64, 128, 512$ for better visualization. Subfigures (a), (b), and (c) have a shared legend. (a) Singular value spectrum of $L$. (b) Proportion of total energy captured by the top $k$ singular values, calculated as $\sum_{i=1}^{k} \sigma_i^2(L_k) / \sum_{i=1}^{r} \sigma_i^2(L)$. (c) Frobenius norm of the dropped matrix $\|L - L_i\|_F$. (d) $\|L - L_i\|_F$ at different number of iterations $t$. Here $T = 50$ and $k = 512$ and the vertical dashed line at $i = 512$ indicates the target rank.

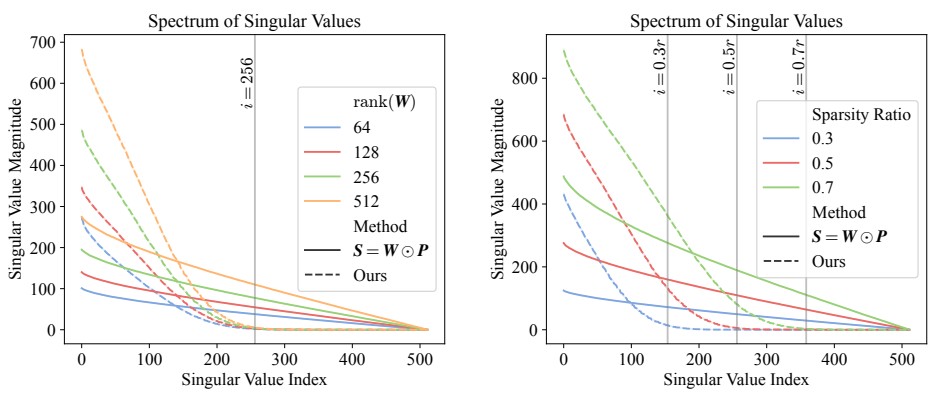

(a) Spectrum of $L$ across various ranks of $W$.

(b) Spectrum of $L$ across various sparsity levels.

Figure 5: Singular value spectrum analysis of $L$ across different conditions. We compare our method with the zero-shot SVD method $L = W \odot (1 - P)$. (a) This subfigure shows the spectrum of $L$ across different ranks of the original weight matrix $W$ (64, 128, 256, 512). (b) This subfigure shows the spectrum of $L$ across different sparsity levels (0.3, 0.5, 0.7) of magnitude pruning.

## C. Experiments with Llama-3.1-8B

We also conducted additional experiments with more recent frontier models to verify our method's effectiveness on state-of-the-art architectures. Specifically, we used Llama-3.1-8B, which represents a significant improvement over the Llama-2 series used in our main experiments, particularly in reasoning and mathematical abilities.

Table 4: Evaluation results on Llama-3.1-8B across various benchmarks.

| Model | HellaSwag | WinoGrande | ARC-e | TruthfulQA | GSM8K | ARC-c | MMLU |
|---|---|---|---|---|---|---|---|
| Dense baseline | 78.9 | 73.6 | 81.1 | 45.2 | 49.8 | 53.4 | 63.5 |
| *Pruning Method* | | | | | | | |
| Magnitude 50% | 56.4 | 57.6 | 56.7 | **42.9** | 1.3 | 35.8 | 35.3 |
| Magnitude 50% with Ours | **66.8** | **67.8** | **68.7** | 38.9 | **6.5** | **42.6** | **45.7** |

As shown in Table 4, our method significantly improves the performance of the magnitude-pruned model across most benchmarks. Notably, on GSM8K, which tests mathematical reasoning abilities, the dense model achieves 49.8% accuracy, while magnitude pruning at 50% sparsity drastically reduces this to 1.3%. Our low-rank refinement method recovers the performance to 6.5%, representing a substantial improvement over basic magnitude pruning. These results demonstrate that our approach is effective even on frontier models with enhanced reasoning capabilities. While there remains a performance gap compared to the dense model, especially on complex reasoning tasks like GSM8K, our method consistently provides substantial recovery of capabilities lost through pruning.

## D. Guidelines for Hyperparameter Selection

The proposed algorithm introduces two key hyperparameters: the target rank $k$ and the number of iterations $T$. For the target rank $k$, we consistently use $k = 128$ across our experiments, which offers a good balance between performance improvement and computational overhead. As shown in Figure 2, larger values of $k$ (e.g., 512) allow the method to retain more information but with diminishing returns relative to the increased computational cost. The optimal choice of $k$ should consider both the model size and the desired performance-efficiency trade-off. Regarding the number of iterations $T$, the primary computational bottleneck is the SVD computation in each iteration, with a time complexity of $O(Tmn^2)$ for a weight matrix $\boldsymbol{W} \in \mathbb{R}^{m \times n}$ (assuming $m \geq n$). We find that $T = 50$ provides sufficient convergence while keeping computational costs reasonable. As demonstrated in Figure 4(d) in Appendix B, further iterations yield diminishing returns in terms of error reduction, a behavior also supported by Theorem 4.5.

## E. End-to-End Inference Acceleration and Memory Usage

In this section, we present a comprehensive analysis of the end-to-end inference acceleration achieved by our method compared to both dense and sparse models.

### E.1. Evaluation without Sparse Matrix Formats

For a fair comparison with the dense baseline model, we first took a conservative approach in our evaluation. We stored all matrices in their full dense format and kept all zero elements (the weight matrices as `torch.Tensor` objects), without leveraging any sparse matrix formats or hardware-specific optimizations for acceleration. Table 5 shows the parameter count and evaluation time for the complete ARC-Challenge benchmark across different model configurations.

As shown in the table, without specialized formats, our method maintains computational efficiency comparable to the dense baseline, with only a marginal increase in relative time ($\sim$1.06$\times$) for the largest model configuration.

### E.2. Evaluation with Sparse Matrix Formats and Hardware Acceleration

To fully exploit the potential efficiency gains from our method, we also evaluated performance when using sparse matrix formats and hardware acceleration. We converted the weight matrices to their sparse format using `torch.sparse.to_sparse_semi_structured` to create `torch.sparse.SparseSemiStructuredTensor` objects. Table 6 compares the memory usage of our proposed method with and without sparse matrix formats.

Table 5: Parameter count and evaluation time for the complete ARC-Challenge benchmark (without sparse matrix formats).

| Model Configuration | Non-Zero Parameter Count | Parameter Count | ARC-C | Relative Time |
|---|---|---|---|---|
| *Llama-2-7B* | | | | |
| Dense baseline | | | 50.5s | 1.0x |
| Magnitude 50% sparse | 3.5B | 6.7B | 50.7s | ∼1.0x |
| Proposed Method (k=128, 50%) | 3.8B | 7.1B | 50.5s | ∼1.0x |
| *Llama-2-13B* | | | | |
| Dense baseline | 13.0B | 13.0B | 73.2s | 1.0× |
| Magnitude 2:4 sparse | 6.7B | 13.0B | 73.9s | ∼1.01× |
| Proposed Method (k=128, 2:4) | 7.2B | 13.5B | 77.9s | ∼1.06× |

Table 6: A comparison of memory usage with and without end-to-end inference acceleration.

| Model Configuration | Memory Usage | Relative Memory Usage |
|---|---|---|
| *Llama-2-13B* | | |
| Proposed Method (k=128, 2:4), dense format | 2×14.6GB | 1.0x |
| Proposed Method (k=128, 2:4), sparse format | 2×9.1GB | ∼0.62x |

When using sparse matrix formats, we observed a significant reduction in memory usage (approximately 38% reduction). The actual inference speedup varies based on factors like batch size and input sequence length. With hardware-accelerated N:M structured pruning (on NVIDIA Ampere GPUs and newer), we observed approximately ∼1.1× faster inference in wall-clock time compared to dense models. It's important to note that for unstructured pruning patterns or on GPUs without dedicated sparse acceleration support, pruning can actually result in slower inference times compared to dense computation.

