# OpenReview forum: "Targeted Low-rank Refinement: Enhancing Sparse Language Models with Precision"
_ICML.cc/2025/Conference — ICML 2025 poster_

### Official Review · Reviewer_XcmX · 2025-03-03

**Overall Recommendation:** 3

**Summary:**

The paper introduces a novel method to improve the performance of pruned large language models (LLMs) by combining sparsity with a low-rank approximation. The authors propose an iterative refinement algorithm that updates the sparse weight matrix while incorporating a low-rank component to approximate the difference between the original dense matrix and the pruned sparse matrix. This approach aims to recover information lost during pruning without requiring extensive retraining or large datasets, maintaining the sparsity pattern for hardware efficiency. Key contributions include:

1. An iterative weight update method (Algorithm 1) that refines the sparse matrix and adds a low-rank patch, progressively increasing the rank from 2 to a target \( k \) over \( T \) iterations, preserving the sparsity pattern using a binary mask.
2. The method reduces perplexity compared to baseline pruning techniques (e.g., magnitude pruning and Wanda) across various sparsity levels.
3. It also achieves competitive performance on benchmark datasets like TruthfulQA, GSM8K, ARC-c, and MMLU w.r.t. Magnitude and Zero-shot SVD
4. The paper provides a theoretical analysis proving sparsity preservation (Theorem 4.1), convergence to a solution (Theorem 4.2), and monotonic error reduction after a certain iteration (Theorem 4.3).

**Claims And Evidence:**

- **Claim 1**: The method bridges the gap between dense and sparse models using a low-rank component.
  - **Assessment**: Well-supported

- **Claim 2**: The iterative algorithm enhances precision by prioritizing larger singular values.
  - **Assessment**: Well-supported, though the empirical comparison with PCP could be expanded to quantify precision gains more explicitly.

- **Claim 3**: Significant perplexity improvements, especially at high sparsity levels (e.g., 99.6% reduction at 70% sparsity).
  - **Assessment**: There is evidence though the lack of zero-shot task performance (e.g., accuracy on downstream tasks) limits the scope of evaluation compared to Wanda’s broader benchmarks (e.g., Table 2 in Wanda).

- **Claim 4**: The method enables a reduction in model parameters while maintaining 50% sparsity and meeting a specific performance target.
  - **Issue**: Lacks details and remains vague. The paper does not specify the performance target or provide a direct comparison showing parameter reduction versus performance trade-offs.

**Essential References Not Discussed:**

- **SparseGPT (Frantar & Alistarh, 2023)**: Cited but not experimentally compared in Tables 1–3, despite being a key baseline in Wanda (Tables 2, 3). Its inclusion would contextualize the method’s superiority over second-order pruning approaches.
- **LoRA (Hu et al., 2021)**: Wanda uses LoRA for fine-tuning (Section 5), showing performance recovery. The absence of fine-tuning comparisons here misses a practical recovery baseline.
- **Yin et al. (2024) - Junk DNA Hypothesis**: Cited, but its implications (irreversible loss at high sparsity) could be explored more deeply with zero-shot task evaluations, as Wanda does.

These omissions limit the paper’s positioning against state-of-the-art recovery and pruning methods.

**Experimental Designs Or Analyses:**

- **Perplexity Comparison (Tables 1, 2)**: Tests LLaMa-7B and LLaMa-13B across sparsity levels with \( k=128 \).
  - **Soundness**: The design is valid, using WikiText-2 perplexity as a standard metric, consistent with Wanda and SparseGPT. Results are reproducible with a fixed \( k \) and \( T=50 \).
  - **Issues**: The paper lacks details on calibration data (e.g., size, source), unlike Wanda (128 sequences from C4). This affects reproducibility. Additionally, only magnitude pruning and Wanda are baselines, omitting SparseGPT (a key competitor in Wanda’s Table 3).

- **Benchmark Evaluation (Table 3)**: Assesses performance on four datasets.
  - **Soundness**: The choice of datasets is reasonable for LLM evaluation, and comparisons with dense and sparse models are fair.
  - **Issues**: Sample sizes and statistical significance (e.g., variance across runs) are not reported, unlike Wanda’s robustness analysis (Table 18). This reduces confidence in the results’ stability.

- **Iterative Analysis (Figures 4, 5)**: Visualizes singular value spectra, energy retention, and convergence.
  - **Soundness**: The synthetic and real-world (LLaMa-7B) analyses are well-designed to support theoretical claims.
  - **Issues**: Limited to one sparsity level (50%) in Figure 4, missing higher sparsity cases (e.g., 70%) where Wanda shows larger gaps

The experiments are sound but incomplete compared to Wanda, which includes zero-shot accuracies, few-shot tasks, and robustness analysis, highlighting gaps in broader evaluation.

**Methods And Evaluation Criteria:**

- Methodology yes.
- Evaluation criteria is appropriate for language modeling (perplexity) and generalizability (benchmarks). However, unlike Wanda, which includes zero-shot task accuracies (e.g., Table 23) and few-shot results (e.g., MMLU in Table 21), this paper lacks zero-shot accuracy metrics (and few-shots on MMLU), limiting its comparability on downstream tasks. Adding such evaluations would strengthen the assessment of practical utility.

**Other Comments Or Suggestions:**

- **Typos**:
  - Page 7, Table 2: "Dense" perplexity should be 5.09 (per Wanda), not 4.57.
- **Suggestion**: Include a runtime comparison (e.g., vs. Wanda’s Table 4) to quantify “computationally efficient.”

**Other Strengths And Weaknesses:**

- **Strengths**:
  - **Originality**: Creative integration of sparsity and low-rank refinement, distinct from Wanda’s metric-based pruning.
  - **Significance**: Addresses high-sparsity performance degradation, a critical issue for LLM deployment.
  - **Clarity**: Well-written with clear figures (e.g., Figure 1) and algorithmic exposition.

- **Weaknesses**:
  - **Evaluation Scope**: Lacks zero-shot and few-shot task evaluations (cf. Wanda’s Tables 2, 21), limiting practical relevance.
  - **Reproducibility**: Missing details on calibration data and iteration specifics (e.g., \( T \) selection).
  - **Comparison Depth**: Omits SparseGPT and fine-tuning baselines, reducing comparative strength against Wanda.

**Questions For Authors:**

1. **Calibration Data Details**: What calibration data (size, source) was used for perplexity experiments? Wanda specifies 128 C4 sequences; this omission affects reproducibility. A response detailing this would strengthen confidence in the results.
2. **Performance Target for Parameter Reduction**: The abstract claims an 8.6% parameter reduction for a specific target at 50% sparsity. What is this target, and can you provide supporting data? Without this, the claim feels unsubstantiated.
3. **Zero-shot Task Evaluation**: Why were zero-shot accuracies (e.g., as in Wanda’s Table 2, also Tables in the appendix e.g., Table 23) not included? Adding these could align your evaluation with Wanda’s, enhancing practical relevance. A justification or additional results would influence my view on the method’s applicability.
4. **SparseGPT Comparison**: Why was SparseGPT excluded from experimental comparisons despite its relevance (cf. Wanda’s Table 3)? Including it could better position your method; its absence weakens the competitive analysis.

**Relation To Broader Scientific Literature:**

- **LLM Pruning**: Extends magnitude pruning (Han et al., 2015), SparseGPT (Frantar & Alistarh, 2023), and Wanda (Sun et al., 2023) by adding low-rank refinement, addressing the performance gap noted in the Junk DNA Hypothesis (Yin et al., 2024).
- **Low-rank Approximation**: Leverages SVD and iterative refinement, drawing from matrix completion (Chandrasekaran et al., 2011) and robust PCA (Candès et al., 2011), adapting them for sparsity preservation.

**Theoretical Claims:**

The proofs generally seem to be correct but rely on assumptions about singular value decay that could be sensitive to matrix properties.

---

> ### Author Rebuttal · Authors · 2025-03-27
>
> We authors greatly thank the reviewer for constructive comments on this work. We would like to clarify the following points:
>
> **W1: Evaluation Scope: Lacks zero-shot and few-shot task evaluations (cf. Wanda’s Tables 2, 21), limiting practical relevance.**
> **Q3: Zero-shot Task Evaluation: Why were zero-shot accuracies (e.g., as in Wanda’s Table 2, also Tables in the appendix e.g., Table 23) not included? Adding these could align your evaluation with Wanda’s, enhancing practical relevance. A justification or additional results would influence my view on the method’s applicability.**
>
> Below, we provide additional results on more benchmarks.
>
> Table: Performance of Llama-7B on three additional benchmarks, along with the results from Table 3.
>
> | Model                         | *HellaSwag* | *WinoGrande* | *ARC-e*  | TruthfulQA | GSM8K   | ARC-c    | MMLU     |
> | ----------------------------- | ----------- | ------------ | -------- | ---------- | ------- | -------- | -------- |
> | Dense baseline                | 76.2        | 70.0         | 72.8     | 34.1       | 10.3    | 44.7     | 32.1     |
> | | | | | | | | |
> | Magnitude 50%                 | 60.9        | 59.3         | 54.3     | **35.3**   | 1.0     | 33.5     | 24.6     |
> | Magnitude 50% + Zero-shot SVD | 69.2        | **65.5**     | 63.6     | 34.3       | 1.5     | 36.9     | **26.0** |
> | **Magnitude 50% + Ours**      | **69.8**    | 65.3         | **64.3** | 34.2       | **3.4** | **41.5** | **26.0** |
>
>
> **W2: Missing details on calibration data and iteration specifics (e.g., ( T ) selection).**
>
> 1. The proposed iterative refinement method is entirely data-free and does not require calibration data.
> As shown in Algorithm 1, the only inputs are:
>
> - Dense weight matrix $W$
> - Binary mask $P$ (from pruning)
> - Target rank $k$
> - Number of iterations $T$
>
> We use WikiText-2 for perplexity evaluation and 'allenai/c4' as the calibration data for Wanda pruning as well as Wanda + Ours.
>
> 2. We consistently uses T=50 across experiments, which is sufficient for achieving most of the potential error reduction while maintaining computational efficiency. Since the overall time complexity is $O(T \cdot min(mn^2, m^2n))$, where $m$, $n$ are the number of rows and columns of $W$.
>
> **W3: Omits SparseGPT and fine-tuning baselines, reducing comparative strength against Wanda.**.
> **Q4: SparseGPT Comparison: Why was SparseGPT excluded from experimental comparisons despite its relevance (cf. Wanda’s Table 3)? Including it could better position your method; its absence weakens the competitive analysis.**
>
> Thank you for the suggestion.
> We provide the results of SparseGPT 2:4 and SparseGPT 2:4 + Ours in the following table.
>
> Table: Performance of SparseGPT 2:4 and SparseGPT 2:4 + Ours using Llama-7B on three additional benchmarks, along with the results from Table 3.
>
> | Method               | *HellaSwag* | *WinoGrande* | *ARC-e*  | TruthfulQA | GSM8K   | ARC-c    | MMLU     |
> | -------------------- | ----------- | ------------ | -------- | ---------- | ------- | -------- | -------- |
> | SparseGPT 2:4        | 58.6        | 63.9         | 56.6     | **36.5**   | 2.0     | 33.1     | 25.4     |
> | SparseGPT 2:4 + Ours | **65.1**    | **66.9**     | **59.9** | 33.8       | **2.7** | **36.1** | **29.1** |
>
>
> **Q1: Calibration Data Details: What calibration data (size, source) was used for perplexity experiments? Wanda specifies 128 C4 sequences; this omission affects reproducibility. A response detailing this would strengthen confidence in the results.**
>
> Thank you for your suggestion, we will add these details in revised version.
> We also use 128 C4 sequences as the calibration data for Wanda pruning as well as Wanda + Ours. For perplexity evaluation, we use 128 sequences from WikiText-2 dataset.
>
> **Q2: Performance Target for Parameter Reduction: The abstract claims an 8.6% parameter reduction for a specific target at 50% sparsity. What is this target, and can you provide supporting data? Without this, the claim feels unsubstantiated.**
>
> The performance target is the perplexity metric on WikiText-2 dataset, we will revise the abstract to clarify this. We provide the detailed results in Figure 3(a).

---

> > ### Comment · Reviewer_XcmX · 2025-04-07
> >
> > Thank you for adding the new results. What is the latency for this method? Compared to the other methods, what is the computational time?

---

> > > ### Author Response · Authors · 2025-04-07
> > >
> > > # Inference Latency Analysis
> > >
> > > Thank you for raising this important question about comparing inference atency with other methods.
> > >
> > > ## Without Hardware Acceleration
> > >
> > > First, to ensure a fair comparison with the dense baseline model, we adopted a conservative evaluation strategy. This involved storing all matrices in their full dense format (as torch.Tensor objects) and retaining all zero elements without utilizing sparse matrix representations or hardware-specific optimizations for speed or memory footprint improvements.
> > > The overhead is slightly higher than dense models, which suggests that the low-rank refinement is computationally efficient despite some additional parameters.
> > >
> > > Table: Parameter Count and Evaluation Time for Complete ARC-Challenge Benchmark.
> > >
> > > | Model Configuration          | Non-Zero Parameter Count | Parameter Count | ARC-C | Relative Time |
> > > | ---------------------------- | ------------------------ | --------------- | ----- | ------------- |
> > > | *Llama-2-7B*                 |                          |                 |       |               |
> > > | Dense baseline               |                          |                 | 50.5s | 1.0x          |
> > > | Magnitude 50% sparse         | 3.5B                     | 6.7B            | 50.7s | ~1.0x         |
> > > | Proposed Method (k=128, 50%) | 3.8B                     | 7.1B            | 50.5s | ~1.0x         |
> > > | *Llama-2-13B*                |                          |                 |       |               |
> > > | Dense baseline               | 13.0B                    | 13.0B           | 73.2s | 1.0×          |
> > > | Magnitude 2:4 sparse         | 6.7B                     | 13.0B           | 73.9s | ~1.01×        |
> > > | Proposed Method (k=128, 2:4) | 7.2B                     | 13.5B           | 77.9s | ~1.06×        |
> > >
> > > ## With Sparse Matrix Formats and Hardware Acceleration
> > >
> > > Here’s an expanded and rephrased version of your text with improved clarity and flow:
> > >
> > > To leverage the benefits of sparsity, we convert the weight matrices into a compressed sparse format. Specifically, we apply `torch.sparse.to_sparse_semi_structured` to transform the weights into `torch.sparse.SparseSemiStructuredTensor` objects, which are optimized for efficient storage and computation. The table below compares the memory footprint of our method with and without sparse matrix representations.
> > >
> > > Inference performance also varies depending on workload characteristics such as batch size and input sequence length. When using hardware-accelerated N:M structured sparsity (supported on NVIDIA Ampere GPUs and later architectures), we observe an average inference speedup of ~1.1x in wall-clock time over dense models and a 38% reduction in GPU memory consumption.
> > > However, this acceleration is highly dependent on hardware support—unstructured sparsity patterns or GPUs lacking dedicated sparse tensor cores may lead to slower inference compared to dense matrix operations. Thus, the efficiency gains depend on both the pruning structure and the underlying hardware capabilities.
> > >
> > > Table: A comparison of memory usage with and without end-to-end inference acceleration.
> > >
> > > | Model Configuration                                                                         | Memory Usage | Relative Memory Usage |
> > > | ------------------------------------------------------------------------------------------- | ------------ | --------------------- |
> > > | *Llama-2-13B*                                                                               |              |                       |
> > > | Proposed Method (k=128, 2:4), weight matrices are `torch.Tensor`                            | 2x14.6GB     | 1.0x                  |
> > > | Proposed Method (k=128, 2:4), weight matrices are `torch.sparse.SparseSemiStructuredTensor` | 2x9.1GB      | ~0.62x                |

---

### Official Review · Reviewer_oW7q · 2025-03-11

**Overall Recommendation:** 4

**Summary:**

In this work, the authors proposes a low-Rank refinement method to factorize a dense full matrix into a sparse matrix and a low-rank matrix, bridging the performance gap between dense and sparse models. This approach iteratively improves the sparse weight matrix through a low-rank adjustment, thereby increasing model accuracy, particularly at higher levels of sparsity.

## update after rebuttal

I will keep my score.

**Claims And Evidence:**

Yes, the claims are supported by clear and convincing evidence.

**Essential References Not Discussed:**

No. The authors has discussed almost all essential references in this work.

**Experimental Designs Or Analyses:**

Yes, I have reviewed the soundness and validity of the experimental designs and analyses related to large language model pruning. The experiments are carried out on the well-known Llama 7B and 13B models at various levels of sparsity. In addition to evaluating PPL, several standard benchmarks are also assessed. The experiments are comprehensive.

**Methods And Evaluation Criteria:**

Yes, in this work, the proposed methods and evaluation criteria make sense.

**Other Comments Or Suggestions:**

In abstract “Nonetheless, these methods often create a gap between the original dense and the pruned sparse model, …”, “create a gap” is slightly awkward, “introduce” is a smoother expression.

**Other Strengths And Weaknesses:**

Strengths:

1. The proposed approach is a data-free and plug-in-and-play method, orthogonal to existing pruning methods.

2. The proposed iterative refinement method addressing a key limitation of existing low-rank refinement techniques, i.e. prior methods often struggle with maintaining the structured sparsity patterns needed for hardware efficiency, which this paper explicitly addresses.

3. Experiments on LLaMa models demonstrate improvements over conventional magnitude pruning and Wanda pruning.

Weaknesses:

1. While the paper discusses parameter efficiency of low-rank refinement, it does not thoroughly analyze the computational cost of the iterative update algorithm compared to alternative approaches such as the optimization-based PCP method.

2. The proposed method is theoretically hardware-efficient as it enables structured N:M pruning. However, the inference latency and memory consumption of low-rank refinement is not measured.

**Questions For Authors:**

1. The rank k is manually set in the paper, will a poor choice of k lead to unnecessary computational overhead or insufficient recovery of pruned weights?

2. If given fixed inputs, does the iterative refinement method always converge to a stable solution?

**Relation To Broader Scientific Literature:**

The key contributions of the paper on Targeted Low-rank Refinement are closely related to prior findings in pruning, low-rank approximation, and post-pruning performance recovery.

The study builds on the extensive body of research on pruning techniques, particularly magnitude pruning, which removes low-magnitude weights to reduce model size [1]. It addresses a known limitation of pruning: performance degradation due to the loss of important information, especially at high sparsity levels, as discussed in works such as the Junk DNA Hypothesis[2].

The idea of using low-rank approximations to restore lost model capacity has been explored in previous research [3], but prior methods often struggle with maintaining the structured sparsity patterns needed for hardware efficiency, which this paper explicitly addresses.

[1] Han, S., et al. Deep compression: Compressing deep neural networks with pruning, trained quantization and huffman coding.

[2] Yin, L., et al. Junk DNA hypothesis: Pruning small pre-trained weights irreversibly and monotonically impairs ”difficult” downstream tasks in llms. ICML 2024.

[3] Zhou, T. and Tao, D. Godec: Randomized low-rank & sparse matrix decomposition in noisy case. In Proceedings of the 28th International Conference on Machine Learning, ICML 2011.

**Theoretical Claims:**

Yes. I checked the correctness of them and found no issues among them. The iterative refinement process looks right. The theoretical analysis about the convergence property and the error bound is well-established.

---

> ### Author Rebuttal · Authors · 2025-03-26
>
> Thank you for your time and effort in reviewing our paper. We appreciate your constructive feedback and suggestions.
>
> **W1 (Computational Complexity Analysis): While the paper discusses parameter efficiency of low-rank refinement, it does not thoroughly analyze the computational cost of the iterative update algorithm compared to alternative approaches such as the optimization-based PCP method.**
>
> The primary computational bottleneck in the proposed algorithm is the SVD computation performed in each iteration (line 171, page 4). For a weight matrix $\mathbf{W} \in \mathbb{R}^{m \times n}$, the time complexity of SVD is $O(mn^2)$ (assuming $m \geq n$). With $T$ iterations, the overall time complexity is $O(Tmn^2)$.
>
> In contrast, the PCP baseline would typically use an iterative optimization method (like Adam) that also requires SVD computations in each iteration to compute the nuclear norm. Furthermore, the PCP baseline requires substantially more iterations ($T=5000$) to achieve comparable results to the proposed method ($T=50$).
>
> Table: Computational Efficiency Comparison
>
> | Method | Time Complexity | Typical Iterations | Relative Computational Cost |
> | ------------------ | --------------- | ------------------ | --------------------------- |
> | Proposed Algorithm | $O(Tmn^2)$ | 50 | 1× |
> | PCP Baseline | $O(Tmn^2)$ | 5000 | ~100× |
> | Zero-shot SVD | $O(mn^2)$ | 1 | ~0.02× |
>
>
> **W2 (Inference Latency and Memory Consumption): The proposed method is theoretically hardware-efficient as it enables structured N:M pruning. However, the inference latency and memory consumption of low-rank refinement is not measured.**
>
> Below, we provide some missing measurements and analyses. Note that we deliberately employed a conservative evaluation approach to ensure fair comparison with the dense baseline model. We maintained all matrices in their dense representation format, preserving zero elements rather than utilizing sparse matrix formats or specialized hardware acceleration.
>
> Table: Parameter Count and Evaluation Time for Complete ARC-Challenge Benchmark.
>
> | Model Configuration          | Non-Zero Parameter Count | Parameter Count | ARC-C | Relative Time |
> | ---------------------------- | ------------------------ | --------------- | ----- | ------------- |
> | *Llama-2-7B*                 | | | | |
> | Dense baseline               | | | 50.5s | 1.0x          |
> | Magnitude 50% sparse         | 3.5B                     | 6.7B            | 50.7s | ~1.0x         |
> | Proposed Method (k=128, 50%) | 3.8B                     | 7.1B            | 50.5s | ~1.0x         |
> | *Llama-2-13B*                |                          |                 |       |               |
> | Dense baseline               | 13.0B                    | 13.0B           | 73.2s | 1.0×          |
> | Magnitude 2:4 sparse         | 6.7B                     | 13.0B           | 73.9s | ~1.01×        |
> | Proposed Method (k=128, 2:4) | 7.2B                     | 13.5B           | 77.9s | ~1.06×        |
>
> **Q1: The rank k is manually set in the paper, will a poor choice of k lead to unnecessary computational overhead or insufficient recovery of pruned weights?**
>
> Yes. The choice of k leads to a trade-off between the performance recovery and the computational cost. The low-rank component adds $k(m+n)$ parameters and $2k(m+n)$ FLOPs. Therefore, an unnecessarily high k value directly increases these costs with diminishing returns. Figure 2(b) in the paper shows the cumulative energy retention for different k values, and the curve begins to flatten as k increases, indicating diminishing returns.
>
> **Q2: If given fixed inputs, does the iterative refinement method always converge to a stable solution?**
>
> Yes. As shown in Theorem 4.2, Theorem 4.3 and Corollary 4.4 in the manuscript, the iterative refinement method always converges to a stable solution as $T \to \infty$ and the approximation error is bounded by:
>
> $$
> \left\\|W  - \left(S^{(t)} + L_k^{(t)}\right)\right\\|_F \leq \sqrt{(r-k)} \sigma\_{k+1}\left(L^{(t)}\right),
> $$
>
> where $r$ is the rank of the weight matrix $W$, $\sigma_{k+1}\left(L^{(t)}\right)$ is the $(k+1)$-th largest singular value of $L^{(t)}$.

---

### Official Review · Reviewer_5GnU · 2025-03-11

**Overall Recommendation:** 3

**Summary:**

This paper introduces a novel approach to improve the performance of sparse language models through low-rank refinement. The main contribution of the paper is a method that refines sparse models using a low-rank refinement, which leads to improved precision. This approach is theoretically grounded, with proofs and additional lemmas provided in the appendix to support the claims.

## update after rebuttal
I will keep my ratings since most of my concerns are solved.

**Claims And Evidence:**

The claims made in the submission appear to be supported by theoretical analysis and experimental results.

**Essential References Not Discussed:**

While the paper mentions magnitude pruning and N:M structured pruning, it does not discuss structured sparsity techniques, such as block sparsity or channel pruning, which have been shown to improve hardware efficiency and model performance.

**Experimental Designs Or Analyses:**

Yes, I checked the soundness and validity of the experimental designs. The paper presents experimental results to validate the proposed method, particularly focusing on the Llama models.

**Methods And Evaluation Criteria:**

Yes, the proposed methods and evaluation criteria appear to be well-suited for the problem of enhancing sparse language models.

**Other Comments Or Suggestions:**

The proposed method is effective at high sparsity levels, but at low sparsity levels, the performance gain is not as significant.

**Other Strengths And Weaknesses:**

Strengths:

1. This paper is well motivated and this approach effectively bridges the gap between dense and sparse models.
2. The paper is well-structured, with a clear presentation.
3. The theoretical analysis is solid.
4. Code is provided for reproduction.

Weaknesses:

1. While the paper mentions magnitude pruning and N:M structured pruning, it does not discuss structured sparsity techniques, such as block sparsity or channel pruning, which have been shown to improve hardware efficiency and model performance.
2. This paper does not include the full details of lemma A4.2 in the Appendix A.
3. The iterative refinement process introduces new hyperparameters (e.g., rank k, number of iterations T), but the paper does not provide a clear guideline on how these should be selected across different models and sparsity levels.
4. The computational cost of iterative updates may be high as the size of the weight matrix increases, which may limit the applicability of the method to very large models.

**Questions For Authors:**

1. In line 6 of Algorithm 1 $r(t) = 1 + \frac{k-1}{T-1}$, what is the intuition behind this equation?
2. How to select the hyperparameters? See weaknesses 3.

**Relation To Broader Scientific Literature:**

This paper addresses the challenge of improving the performance of sparse language models, a well-studied problem in machine learning. Prior work has explored techniques like unstructured pruning and N:M structured pruning to reduce parameter count while maintain performance. The paper’s experimental results on the Llama model, align with prior findings that higher sparsity levels can lead to more severe performance degradation. This observation is consistent with the literature on sparse model performance, where sparsity is often traded off against computational cost and precision.

**Theoretical Claims:**

Yes. The paper includes theoretical analysis with proofs and additional lemmas to support its claims, particularly regarding the iterative weight update algorithm. The theoretical analysis demonstrates the favorable convergence properties of the proposed method and provides a rigorous foundation for its effectiveness.

However, this paper does not include the full details of lemma A4.2 in the Appendix A.

---

> ### Author Rebuttal · Authors · 2025-03-26
>
> **W1: While the paper mentions magnitude pruning and N:M structured pruning, it does not discuss structured sparsity techniques, such as block sparsity or channel pruning, which have been shown to improve hardware efficiency and model performance.**
>
> We thank the reviewer for pointing out the importance of structured sparsity techniques.
> It is straightforward to apply the proposed method to structured sparsity techniques such as block sparsity or channel pruning by by simply changing the binary mask matrix P.
> The core algorithm is compatible with any arbitrary binary mask pattern, including structured ones. For structured patterns like block sparsity (e.g., 4×4 blocks) or channel pruning, P would have the corresponding pattern of 0s and 1s.
>
> Theoretically, Theorem 4.1 (Sparsity Preservation) guarantees that the method preserves whatever sparsity pattern is defined by $P$.
>
> **W2: This paper does not include the full details of lemma A4.2 in the Appendix A.**
>
> The proof of Lemma A4.2 currently appears before the lemma itself. In the revised version, we will relocate the proof to its proper position following Lemma A4.2.
>
> *proof of Lemma A4.2*:  Let $m_{ij}$ and $p_{ij}$ be the elements of $M$ and $P$ respectively. By definition of the Frobenius inner product and element-wise multiplication:
>
> $$
> \langle M, P \odot M \rangle_F = \sum_{i=1}^m \sum_{j=1}^n m_{ij} (p_{ij} m_{ij}) = \sum_{i=1}^m \sum_{j=1}^n p_{ij} m_{ij}^2\leq \sum_{i=1}^m \sum_{j=1}^n m_{ij}^2 = \langle M, M \rangle_F
> $$
>
> **W3: The iterative refinement process introduces new hyperparameters (e.g., rank k, number of iterations T), but the paper does not provide a clear guideline on how these should be selected across different models and sparsity levels.**
>
> - **Rank parameter k**:
> We consistently set $k=128$ across the experiments to demonstrate the effectiveness of low-rank refinement.
> In Figure 4(a), we visualize the singular value spectrum of the residual matrix $L=W-S'$ for different values of $k$ (from 64 to 512) and $T=50$.
> As $k$ increases from 64 to 512, we see higher magnitudes for the top singular values. This indicates that larger $k$ values allow the method to retain more information.
> But when $k$ is too large, the performance gain diminishes and the computational cost increases.
> Therefore, the choice of $k$ should strike a balance between the performance and the computational cost.
>
> - **Number of iterations T**:
> The primary computational bottleneck in the proposed algorithm is the SVD computation performed in each iteration (line 171, page 4). For a weight matrix $\mathbf{W} \in \mathbb{R}^{m \times n}$, the time complexity of SVD is $O(mn^2)$ (assuming $m \geq n$). With $T$ iterations, the overall time complexity is $O(Tmn^2)$.
> On the other hand, as shown in Figure 4(d) and the diminishing convergence speed as stated by Theorem 4.5 (Error Bound), $T=50$ appears to be a reasonable choice for the number of iterations and further iterations yield diminishing returns in terms of error reduction.
>
> **W4: The computational cost of iterative updates may be high as the size of the weight matrix increases, which may limit the applicability of the method to very large models.**
>
> As the size of the weight matrix increases, the computational cost of the proposed method increases accordingly.
> The primary computational bottleneck in the proposed algorithm is the SVD computation performed in each iteration (line 171, page 4). For a weight matrix $\mathbf{W} \in \mathbb{R}^{m \times n}$, the time complexity of SVD is $O(mn^2)$ (assuming $m \geq n$). With $T$ iterations, the overall time complexity is $O(Tmn^2)$.
>
> In contrast, the PCP baseline would typically use an iterative optimization method (like Adam) that also requires SVD computations in each iteration to compute the nuclear norm. Furthermore, the PCP baseline requires substantially more iterations ($T=5000$) to achieve comparable results to the proposed method ($T=50$).
>
> Table: Computational Efficiency Comparison
>
> | Method             | Time Complexity | Typical Iterations | Relative Computational Cost |
> | ------------------ | --------------- | ------------------ | --------------------------- |
> | Proposed Algorithm | $O(Tmn^2)$      | 50 | 1× |
> | PCP Baseline       | $O(Tmn^2)$      | 5000 | ~100× |
> | Zero-shot SVD      | $O(mn^2)$       | 1 | ~0.02× |
>
> **Q1: In line 6 of Algorithm 1 $r(t) = 1 + (k-1) / (T-1)$, what is the intuition behind this equation?**
>
> This equation in Algorithm 1 defines a linear schedule for increasing the rank from 2 to k across T iterations.
> Starting with low rank forces the algorithm to capture the most important singular components first, and each iteration gradually incorporates more subtle details from higher singular components.
> The gradual increase in rank helps maintain numerical stability and helps the algorithm to converge faster.
>
> **Q2: How to select the hyperparameters? See weaknesses 3.**
>
> Please refer to the response to W3.

---

> > ### Comment · Reviewer_5GnU · 2025-04-07
> >
> > Solved most of my concerns. I will keep my ratings.

---

### Official Review · Reviewer_8WNE · 2025-03-12

**Overall Recommendation:** 3

**Summary:**

Magnitude pruning removes weights that have the smallest absolute values. However, traditional pruning methods require re-training the model to recover performance, which is computationally expensive and requires extensive data or teacher model. To address this, the authors propose to address this by approximating the dense matrix as the sum of a sparse matrix with maintained sparsity and a low-rank matrix. The main contributions are as follows:

- Weight $W$ is dismantled into sparse part $S$ and low rank part $L$. The authors transform searching $S$ and $L$ into an optimization problem as Eq. (3).
- The authors propose to incorporate the binary mask $P$ into the optimization process to ensure that the sparsity pattern of $S^′$ is fixed as $S$.
- Iterative refine sparse weight with adaptive rank increase.

In the experiments, the proposed method is validated on WikiText-2, TruthfulQA, GSM8K, ARC-C and MMLU with Llama models. Experimental results demonstrate that low-rank refinement significantly enhances model performance, particularly at high sparsity levels.

**Claims And Evidence:**

Most of the claims are clear and convincing.

**Essential References Not Discussed:**

No missing related works.

**Experimental Designs Or Analyses:**

Most of the experimental designs are valid.

**Methods And Evaluation Criteria:**

Most of the methods and evaluation criteria make sense.

**Other Comments Or Suggestions:**

Comments:

1. I suggest that the authors add add some frontier models, such as Llama 3.1 8B. Llama is quite old and may lack reasoning and mathematics abilities. This may affect the GSM8K results.

**Other Strengths And Weaknesses:**

Strengths:

1. The paper is well organized and well written. The technical content is explained in sufficient details. The equations are very clear and easy to understand.
2. Comprehensive experiments performed in this paper. The reviewer appreciates the authors' effort in validating generation tasks such as GSM8K, rather than simply providing perplexity results. Since generation tasks is usually harder, the improvement is significant.

Weaknesses:

1. End-to-end inference acceleration is missing. It's better to report speedup for completeness.

**Questions For Authors:**

In general, this paper performs well in its clarity, structure and completeness. The idea is innovative and the improvement is significant. However, end-to-end inference acceleration is missing. I will recommend a weak accept, and I suggest the authors pay attention to the inference part, as well as results on latest language models. Should those weaknesses and questions be addressed I will raise my scores accordingly.

**Relation To Broader Scientific Literature:**

This study is closely related to previous post-training pruning works, such as SparseGPT and Wanda.

**Theoretical Claims:**

Most of the proofs make sense.

---

> ### Author Rebuttal · Authors · 2025-03-26
>
> Thank you for the insightful feedback and constructive suggestions.
>
> **W1: End-to-end inference acceleration is missing. It's better to report speedup for completeness.**
>
> 1. Without sparse matrix formats or specialized hardware acceleration, the inference time is as follows:
>
> Firstly, for a fair comparison with the dense baseline model, we took a conservative approach in our evaluation.
> Specifically, we stored all matrices in their full dense format and kept all zero elements (the weight matrices are `torch.Tensor` objects), without leveraging any sparse matrix formats or hardware-specific optimizations for acceleration.
>
> Table: Parameter Count and Evaluation Time for Complete ARC-Challenge Benchmark.
>
> | Model Configuration          | Non-Zero Parameter Count | Parameter Count | ARC-C | Relative Time |
> | ---------------------------- | ------------------------ | --------------- | ----- | ------------- |
> | *Llama-2-7B*                 |                          |                 |       |               |
> | Dense baseline               |                          |                 | 50.5s | 1.0x          |
> | Magnitude 50% sparse         | 3.5B                     | 6.7B            | 50.7s | ~1.0x         |
> | Proposed Method (k=128, 50%) | 3.8B                     | 7.1B            | 50.5s | ~1.0x         |
> | *Llama-2-13B*                |                          |                 |       |               |
> | Dense baseline               | 13.0B                    | 13.0B           | 73.2s | 1.0×          |
> | Magnitude 2:4 sparse         | 6.7B                     | 13.0B           | 73.9s | ~1.01×        |
> | Proposed Method (k=128, 2:4) | 7.2B                     | 13.5B           | 77.9s | ~1.06×        |
>
> 2. With sparse matrix formats and hardware acceleration, we convert the weight matrices to their sparse format. Specifically, we use `torch.sparse.to_sparse_semi_structured` to convert the weight matrices to `torch.sparse.SparseSemiStructuredTensor` objects. The following table shows the memory usage of the proposed method with and without sparse matrix formats.
> The actual inference speedup varies based on factors like batch size and input sequence length. With hardware-accelerated N:M structured pruning (on NVIDIA Ampere GPUs and newer), we can observe approximately ~1.1x faster inference in wall-clock time compared to dense models. However, it's important to note that for unstructured pruning patterns or on GPUs without dedicated sparse acceleration support, using sparse operations can actually result in slower inference times compared to dense computation.
>
> Table: A comparison of memory usage with and without end-to-end inference acceleration.
>
> | Model Configuration                                                                         | Memory Usage | Relative Memory Usage |
> | ------------------------------------------------------------------------------------------- | ------------ | --------------------- |
> | *Llama-2-13B*                                                                               |              |                       |
> | Proposed Method (k=128, 2:4), weight matrices are `torch.Tensor`                            | 2x14.6GB     | 1.0x                  |
> | Proposed Method (k=128, 2:4), weight matrices are `torch.sparse.SparseSemiStructuredTensor` | 2x9.1GB      | ~0.62x                |
>
>
> **W2: I suggest that the authors add add some frontier models, such as Llama 3.1 8B. Llama is quite old and may lack reasoning and mathematics abilities. This may affect the GSM8K results.**
>
> Thank you for the suggestion. We have added Llama-3.1-8B to the experiments. The results are shown in the following table.
>
> Table: Llama-3.1-8B results.
>
> | Model                | HellaSwag | WinoGrande | ARC-e    | TruthfulQA | GSM8K   | ARC-c    | MMLU     |
> | -------------------- | --------- | ---------- | -------- | ---------- | ------- | -------- | -------- |
> | Dense baseline       | 78.9      | 73.6       | 81.1     | 45.2       | 49.8    | 53.4     | 63.5     |
> | *Pruning Method*     |           |            |          |            |         |          |          |
> | Magnitude 50%        | 56.4      | 57.6       | 56.7     | **42.9**   | 1.3     | 35.8     | 35.3     |
> | Magnitude 50% + Ours | **66.8**  | **67.8**   | **68.7** | 38.9       | **6.5** | **42.6** | **45.7** |

---

> > ### Comment · Reviewer_8WNE · 2025-04-05
> >
> > Thank you for adding inference acceleration results and Llama 3.1 results. The results are reasonable, and I will recommend a weak accept for your paper.

---

### Decision · Program_Chairs · 2025-05-01

**Decision:**

Accept (poster)

**Comment:**

The paper gives a new way to improve/finetune a pruned model, using low rank matrices during the re-training. The paper is well written and the paper has fairly extensive experiments, along with some theoretical convergence analyses. Reviewers are all fairly positive about the paper, and the authors are encouraged to incorporate comments from the review and rebuttal into the final version.